# ARE LARGE LANGUAGE MODELS POST HOC EXPLAINERS?

## ABSTRACT

Large Language Models (LLMs) are increasingly used as powerful tools for a plethora of natural language processing (NLP) applications. A recent innovation, in-context learning (ICL), enables LLMs to learn new tasks by supplying a few examples in the prompt during inference time, thereby eliminating the need for model fine-tuning. While LLMs have been utilized in several applications, their applicability in explaining the behavior of other models remains relatively unexplored. Despite the growing number of new explanation techniques, many require white-box access to the model and/or are computationally expensive, highlighting a need for next-generation post hoc explainers. In this work, we present the first framework to study the effectiveness of LLMs in explaining other predictive models. More specifically, we propose a novel framework encompassing multiple prompting strategies: i) Perturbation-based ICL, ii) Prediction-based ICL, iii) Instruction-based ICL, and iv) Explanation-based ICL, with varying levels of information about the underlying ML model and the local neighborhood of the test sample. We conduct extensive experiments with real-world benchmark datasets to demonstrate that LLM-generated explanations perform on par with state-of-the-art post hoc explainers using their ability to leverage ICL examples and their internal knowledge in generating model explanations. On average, across four datasets and two ML models, we observe that LLMs identify the most important feature with 72.19% accuracy, indicating promising avenues for further research into LLM-based explanation frameworks within explainable artificial intelligence (XAI).

## 1 INTRODUCTION

Over the past decade, machine learning (ML) models have become ubiquitous across various industries and applications. With their increasing use in critical applications (*e.g.,* healthcare, financial systems, and crime forecasting), it becomes essential to ensure that ML developers and practitioners understand and trust their decisions. To this end, several approaches (Ribeiro et al., 2016; 2018; Smilkov et al., 2017; Sundararajan et al., 2017; Lundberg & Lee, 2017; Shrikumar et al., 2017) have been proposed in explainable artificial intelligence (XAI) literature to generate explanations for understanding model predictions. However, these explanation methods are highly sensitive to changes in their hyperparameters (Yeh et al., 2019; Bansal et al., 2020), require access to the underlying black-box ML model (Lundberg & Lee, 2017; Ribeiro et al., 2016), and/or are often computationally expensive (Situ et al., 2021), thus impeding reproducibility and the trust of relevant stakeholders.

More recently, generative models such as Large Language Models (LLMs) (Radford et al., 2017) have steered ML research into new directions and shown exceptional capabilities, allowing them to surpass state-of-the-art models at complex tasks like machine learning translation (Hendy et al., 2023), language understanding (Brown et al., 2020), commonsense reasoning (Wei et al., 2022b; Krishna et al., 2023), and coding tasks (Bubeck et al., 2023). However, there is very little work on systematically analyzing the reliability of LLMs as explanation methods. While recent research has used LLMs to explain what patterns in a text cause a neuron to activate, they simply explain correlations between the network input and specific neurons and do not explain what causes model behavior at a mechanistic level (Bills et al., 2023). Thus, the ability of LLMs to act as reliable explainers and improve the understanding of ML models lacks sufficient exploration.

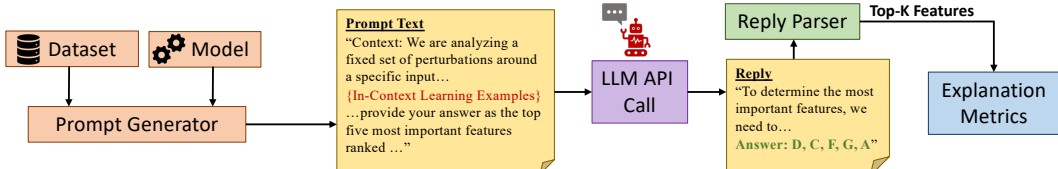

Figure 1: **Overview of our framework.** Given a dataset and model to explain, we provide 1) different prompting strategies to generate explanations using LLMs, 2) functions to parse LLM-based explanations, 3) utility functions to support new LLMs, and 4) diverse performance metrics to evaluate the faithfulness of explanations.

**Present work.** In this work, we present the first framework to study the effectiveness of LLMs in explaining other predictive models (see Fig. 1). More specifically, we introduce four broad prompting strategies — Perturbation-based ICL, Prediction-based ICL, Instruction-based ICL, and Explanation-based ICL — for generating post hoc explanations using LLMs. Our first three strategies entail providing local neighborhood samples and labels of a given instance whose prediction we want to explain, before asking an LLM to identify features that are key drivers in the model's predictions. In our last approach, we leverage the in-context learning (ICL) (Liu et al., 2023b) behavior of LLMs by providing a small set of instances and their corresponding explanations (output by state-of-the-art post hoc explanation methods) as input to an LLM and ask it to generate feature importance-based explanations for new samples. We also explore different prompting and design choices, such as increasing the level of information in each, to generate more faithful explanations using LLMs.

We conduct extensive experimentation with four benchmark datasets, two black-box models, and two GPT models to analyze the efficacy of our proposed framework. Our empirical studies reveal the following key findings. 1) LLMs, on average, accurately identify the most important feature (top-$k$=1) with 72.19% accuracy across different datasets, with performance drop for larger values of top-$k$ features. 2) LLMs can mimic the behavior of six state-of-the-art post hoc explanation methods using the proposed Explanation-based ICL prompting strategy and only four ICL samples. On average, LLMs behave as post hoc explainers by providing explanations that are on par with existing methods, such as LIME and gradient-based methods, in terms of their faithfulness. 3) LLMs struggle to retrieve relevant information from longer prompts, resulting in a decrease in the faithfulness of the explanations generated using a large set of ICL samples. 4) Our proposed framework paves the way for a new paradigm in XAI research, where LLMs can aid in explaining black-box model predictions.

## 2   RELATED WORKS

Our work lies at the intersection of post hoc explanations, large language models, and in-context learning, which we discuss below.

**Post Hoc Explanations.** The task of understanding model predictions has become increasingly intricate with the growing popularity of complex ML models (Doshi-Velez & Kim, 2017) due to their inherent black box nature, which makes it difficult to interpret their internal reasoning. To this end, a plethora of feature attribution methods (commonly referred to as post hoc explanation methods) have been proposed to provide explanations for these models' predictions. These explanations are predominantly presented in the form of feature attributions, which highlight the importance of each input feature on the model's prediction. Broadly, post hoc explainers can be divided into perturbation-based and gradient-based methods. While perturbation-based methods (Ribeiro et al., 2016; Lundberg & Lee, 2017; Zeiler & Fergus, 2014) leverage perturbations of the given instance to construct an interpretable approximation of the black-box model behavior, gradient-based methods (Smilkov et al., 2017; Sundararajan et al., 2017) leverage gradients *w.r.t.* the given instance to explain model predictions. In this work, we primarily focus on state-of-the-art local post hoc explainers, *i.e.,* methods explaining individual feature importance for model predictions of individual instances.

**Large Language Models.** LLMs have seen exponential growth in recent years, both in terms of their size and the complexity of tasks they can perform (Radford et al., 2017). Recent advances in LLMs like GPT-4 (OpenAI), Bard (Google), Claude-2 (Anthropic) and Llama-2 (Meta) are changing the paradigm of NLP research and have led to their widespread use across applications spanning machine translation (Vaswani et al., 2017), question-answering (Brown et al., 2020), text genera-

tion (Radford et al., 2017), and medical data records (Lee et al., 2020; Alsentzer et al., 2019). In this work, we, for the first time, explore the use of LLMs in explaining other predictive models.

**In-context Learning.** While the high performance and generalization capabilities have led to highly effective language models for numerous tasks (Wei et al., 2022a), they have also increased the models' parameter sizes and the computational costs for additional fine-tuning on new downstream tasks. To alleviate this, recent works have introduced *in-context learning* (ICL), which allows an LLM to perform well on new tasks by simply using a few task samples in the prompt (Liu et al., 2023b). Despite their effectiveness in enhancing the performance of LLMs, these methods have not been thoroughly explored for their potential to generate post-hoc explanations. In this work, we investigate the utility of LLMs in generating post hoc explanations by leveraging their in-context learning abilities.

## 3   OUR FRAMEWORK

Next, we describe our framework that aims to generate explanations using LLMs. To achieve this goal, we outline four distinct prompting strategies — *Perturbation-based ICL* (Sec. 3.1), *Prediction-based ICL* (Sec. 3.2), *Instruction-based ICL* (Sec. 3.3), and *Explanation-based ICL* (Sec. 3.4).

**Notation.** Let $f : \mathbb{R}^d \rightarrow [0, 1]$ denote a black-box ML model that takes an input $\mathbf{x} \in \mathbb{R}^d$ and returns the probability of $\mathbf{x}$ belonging to a class $c \in C$ and the predicted label $y$. Following previous XAI works (Ribeiro et al., 2016; Smilkov et al., 2017), we randomly sample points from the local neighborhood $\mathcal{N}_x$ of the given input $\mathbf{x}$ to generate explanations, where $\mathcal{N}_x = \mathcal{N}(\mathbf{x}, \sigma^2)$ denotes the neighborhood of perturbations around $\mathbf{x}$ using a Normal distribution with mean 0 and variance $\sigma^2$.

### 3.1   PERTURBATION-BASED ICL

In the Perturbation-based ICL prompting strategy, we use an LLM to explain $f$, trained on tabular data, by querying the LLM to identify the top-$k$ most important features in determining the output of $f$ in a rank-ordered manner. To tackle this, we sample input-output pairs from the neighborhood $\mathcal{N}_x$ of $\mathbf{x}$ and generate their respective strings following a serialization template; for instance, a perturbed sample's feature vector $\mathbf{x}' = [0.058, 0.632, -0.015, 1.012, -0.022, -0.108]$, belonging to class 0 in the COMPAS dataset, is converted into a natural-language string as:

```
# Serialization template
Input: A = 0.058, B = 0.632, C = -0.015, D = 1.012, E = -0.022, F = -0.108
Output: 0
```

While previous post hoc explainers suggest using a large number of neighborhood samples (Ribeiro et al., 2016; Smilkov et al., 2017), it is impractical to provide all samples from $\mathcal{N}_x$ in the prompt for an LLM due to their constraint on the maximum context length and performance loss when given more information (Liu et al., 2023a). Consequently, we select $n_{\text{ICL}}$ samples from $\mathcal{N}_x$ to use in the LLM's prompt. In the interest of maintaining a neutral and fundamental approach, we employ two primary sampling strategies, both selecting balanced class representation within the neighborhoods defined by $\mathcal{N}_x$. The first strategy selects samples randomly, while the second chooses those with the highest confidence levels, aiding the LLM in generating explanations centered on model certainty.

Given $n_{\text{ICL}}$ input-output pairs from $\mathcal{N}_x$ and the test sample $\mathbf{x}$ to be explained, we add context with respect to the predictive model, dataset, and task description in our prompt to aid the LLM in behaving like a post hoc explanation method. Motivated by the local neighborhood approximation works in XAI, the Perturbation-based ICL prompting strategy presumes that the local behavior of $f$ is a simple linear decision boundary, contrasting with the often globally exhibited complex non-linear decision boundary. Hence, assuming a sufficient number of perturbations in $\mathcal{N}_x$, the LLM is expected to accurately approximate the black box model's behavior and utilize this information to identify the top-$k$ most important features. The final prompt structure is given below, where the *"Context"* provides the LLM with the background of the underlying ML model, the number of features in the dataset, and model predictions, *"Dataset"* denotes the $n_{\text{ICL}}$ instances sampled from the neighborhood $\mathcal{N}_x$ of $\mathbf{x}$, *"Question"* is the task we want our LLM to perform, and *"Instructions"* are the guidelines we want the LLM to follow while generating the output explanations.

> **# Perturbation-based ICL Prompt Template**
> **Context:** *"We have a two-class machine learning model that predicts based on 6 features: ['A', 'B', 'C', 'D', 'E', 'F']. The model has been trained on a dataset and has made the following predictions."*
> **Dataset:**
> *Input: A = -0.158, B = 0.293, C = 0.248, D = 1.130, E = 0.013, F = -0.038*
> *Output: 0*
> *...*
> *Input: A = 0.427, B = 0.016, C = -0.128, D = 0.949, E = 0.035, F = -0.045*
> *Output: 1*
> **Question:** *"Based on the model's predictions and the given dataset, what appears to be the top five most important features in determining the model's prediction?"*
> **Instructions:** *"Think about the question. After explaining your reasoning, provide your answer as the top five most important features ranked from most important to least important, in descending order. Only provide the feature names on the last line. Do not provide any further details on the last line."*

## 3.2 PREDICTION-BASED ICL

Here, we devise Prediction-based ICL, a strategy closer to the traditional ICL prompting style, where the primary objective remains the same — understanding the workings of the black-box model $f$ by identifying the top-$k$ most important features. This strategy positions the LLM to first emulate the role of the black-box model by making predictions, staging it to extract important features that influenced its decision. We follow the perturbation strategy of Sec. 3.1 and construct the Prediction-based ICL prompt using $n_{ICL}$ input-output pairs from $\mathcal{N}_x$. The main difference in the Prediction-based ICL prompting strategy lies in the structuring of the prompt, which is described below:

> **# Prediction-based ICL Prompt Template**
> **Context:** *"We have a two-class machine learning model that predicts based on 6 features: ['A', 'B', 'C', 'D', 'E', 'F']. The model has been trained on a dataset and has made the following predictions."*
> **Dataset:**
> *Input: A = 0.192, B = 0.240, C = 0.118, D = 1.007, E = 0.091, F = 0.025*
> *Output: 0*
> *...*
> *Input: A = 0.709, B = -0.102, C = -0.177, D = 1.056, E = -0.056, F = 0.015*
> *Output: 1*
> *Input: A = 0.565, B = -0.184, C = -0.386, D = 1.003, E = -0.123, F = -0.068*
> *Output:*
> **Question:** *"Based on the model's predictions and the given dataset, estimate the output for the final input. What appears to be the top five most important features in determining the model's prediction?"*
> **Instructions:** *"Think about the question. After explaining your reasoning, provide your answer as the top five most important features ranked from most important to least important, in descending order. Only provide the feature names on the last line. Do not provide any further details on the last line."*

Here, we construct the prompt using the task description followed by the $n_{ICL}$ ICL samples and then ask the LLM to provide the predicted label for the test sample **x** and explain how it generated that label. The primary motivation behind the Prediction-based ICL prompting strategy is to investigate whether the LLM can learn the classification task using the ICL set and, if successful, identify the important features in the process. This approach aligns more closely with the traditional ICL prompting style, offering a different perspective on the problem.

## 3.3 INSTRUCTION-BASED ICL

The Instruction-based prompting transitions from specifying task objectives to providing detailed guidance on the strategy for task execution. Rather than solely instructing the LLM on what the task entails, this strategy delineates how to conduct the given task. The objective remains to understand the workings of the black-box model and identify the top-$k$ most important features. However, in using step-by-step directives, we aim to induce a more structured and consistent analytical process within the LLM to target more faithful explanations. The final prompt structure is as follows:

*# Instruction-based ICL Prompt Template*
**Context:** *"We are analyzing a fixed set of perturbations around a specific input to understand the influence of each feature on the model's output. The dataset below contains the change in features 'A' through 'F' (with negative values denoting a decrease in a feature's value) and the corresponding outputs."*
**Dataset:**
*Change in Input: A: -0.217, B: 0.240, C: 0.114, D: 0.007, E: 0.091, F: 0.025*
*Change in Output: -1*
*. . .*
*Change in Input: A: 0.185, B: -0.185, C: -0.232, D: -0.130, E: -0.020, F: 0.015*
*Change in Output: 0*
**Instructions:** *"For each feature, starting with 'A' and continuing to 'F':*
*1. Analyze the feature in question:*
*a. Compare instances where its changes are positive to where its changes are negative and explain how this difference correlates with the change in output.*
*b. Rate the importance of the feature in determining the output on a scale of 0-100, considering both positive and negative correlations. Ensure to give equal emphasis to both positive and negative correlations and avoid focusing only on absolute values.*
*2. After analyzing the feature, position it in a running rank compared to the features already analyzed. For instance, after analyzing feature 'B', determine its relative importance compared to 'A' and position it accordingly in the rank (e.g., BA or AB). Continue this process until all features from 'A' to 'F' are ranked. Upon completion of all analyses, provide the final rank of features from 'A' to 'F' on the last line. Avoid providing general methodologies or suggesting tools. Justify your findings as you go."*

Here, we provide some general instructions to the LLM for understanding the notion of important features and how to interpret them through the lens of correlation analysis. To achieve this, we instruct LLMs to analyze each feature sequentially and ensure that both positive and negative correlations are equally emphasized. The LLM assigns an importance score for each feature in the given dataset and then positions it in a running rank. This rank is necessary to differentiate features and avoid ties in the LLM's evaluations. The final line ensures that the LLM's responses are strictly analytical, minimizing non-responsiveness or digressions into tool or methodology recommendations.

### 3.4 EXPLANATION-BASED ICL

Recent studies show that LLMs can learn new tasks through ICL, enabling them to excel in new downstream tasks by merely observing a few instances of the task in the prompt. In the Explanation-based ICL prompting strategy, we leverage the ICL capability of LLMs to alleviate the computation complexity of some post hoc explanation methods. In particular, we investigate whether an LLM can mimic the behavior of a post hoc explainer by looking at a few input, output, and explanation examples. We generate explanations for a given test sample $\mathbf{x}$ using LLMs by utilizing the ICL framework and supplying $n_{ICL}$ input, output, and explanation examples to the LLM, where the explanations in the ICL can be generated using any post hoc explanation method. For constructing the ICL set, we randomly select $n_{ICL}$ input instances $\mathbf{X}_{ICL}$ from the ICL split of the dataset and generate their predicted labels $\mathbf{y}_{ICL}$ using model $f$. Next, we generate explanations $\mathbf{E}_{ICL}$ for samples ($\mathbf{X}_{ICL}$, $\mathbf{y}_{ICL}$) using any post hoc explainer. Using the above input, output, and explanation samples, we construct a prompt by concatenating each pair as follows:

*# Explanation-based ICL Prompt Template*
*Input: A = 0.172, B = 0.000, C = 0.000, D = 1.000, E = 0.000, F = 0.000*
*Output: 1*
*Explanation: A,C,B,F,D,E*
*. . .*
*Input: A = 0.052, B = 0.053, C = 0.073, D = 0.000, E = 0.000, F = 1.000*
*Output: 0*
*Explanation: A,B,C,E,F,D*
*Input: A = 0.180, B = 0.222, C = 0.002, D = 0.000, E = 0.000, F = 1.000*
*Output: 0*
*Explanation:*

Using the Explanation-based ICL prompting strategy, we aim to investigate the learning capability of LLMs such that they can generate faithful explanations by examining the $n_{ICL}$ demonstration pairs of inputs, outputs, and explanations generated by state-of-the-art post hoc explainer.

# 4    EXPERIMENTS

Next, we evaluate the effectiveness of LLMs as post hoc explainers. More specifically, our experimental analysis focuses on the following questions: Q1) Can LLMs generate faithful post hoc explanations? Q2) Do LLM-Augmented post hoc explainers achieve similar faithfulness vs. their vanilla counterpart? Q3) Are LLMs better than state-of-the-art post hoc explainers at identifying the most important feature? Q4) Is GPT-4 a better explainer than GPT-3.5? Q5) Are changes to the LLM's prompting strategy necessary for generating faithful explanations?

## 4.1    DATASETS AND EXPERIMENTAL SETUP

We first describe the datasets and models used to study the reliability of LLMs as post hoc explainers and then outline the experimental setup.

**Datasets.** Following previous LLM works (Hegselmann et al., 2023), we performed analysis on four real-world tabular datasets: **Blood** (Yeh et al., 2009) having four features, **Recidivism** (ProPublica) having six features, **Adult** (Kaggle) having 13 features, and **Default Credit** (UCI) having 10 features. The datasets come with a random train-test split, and we further subdivide the train set, allocating 80% for training and the remaining 20% for ICL sample selection, as detailed in Sec. 3.4. We use a random set of 100 samples from the test split to generate explanations for all of our experiments.

**Predictive Models.** We consider two ML models with varying complexity in our experiments: i) Logistic Regression (LR) and ii) Artificial Neural Networks (ANN). We use PyTorch (Paszke et al., 2019) to implement the ANNs with the following combination of hidden layers: one layer of size 16 for the LR model; and three layers of size 64, 32, and 16 for the ANN, using RELU for the hidden layers and SOFTMAX for the output (see Table 1 for predictive performances of these models).

**Large Language Model.** We consider GPT-3.5 and GPT-4 as language models for all experiments.

**Baseline Explanation Methods.** We use six post hoc explainers as baselines to investigate the effectiveness of explanations generated using LLMs: LIME (Ribeiro et al., 2016), SHAP (Lundberg & Lee, 2017), Vanilla Gradients (Zeiler & Fergus, 2014), SmoothGrad (Smilkov et al., 2017), Integrated Gradients (Sundararajan et al., 2017), and Gradient x Input (ITG) (Shrikumar et al., 2017).

**Performance Metrics.** We employ four distinct metrics to measure the faithfulness of an explanation. To quantify the faithfulness of an explanation where there exists a ground-truth top-$k$ explanation for each test input (*i.e.,* LR model coefficients), we use the Feature Agreement (FA) and Rank Agreement (RA) metrics introduced in Krishna et al. (2022), which compares the LLM's top-$k$ directly with the model's ground-truth. The FA and RA metrics range from $[0, 1]$, where 0 means no agreement and 1 means full agreement. However, in the absence of a top-$k$ ground-truth explanation (as is the case with ANNs), we use the Prediction Gap on Important feature perturbation (PGI) and the Prediction Gap on Unimportant feature perturbation (PGU) metrics from OpenXAI (Agarwal et al., 2022). While PGI measures the change in prediction probability that results from perturbing the features deemed as influential, PGU examines the impact of perturbing unimportant features. Here, the perturbations are generated using Gaussian noise $\mathcal{N}(0, \sigma^2)$.

**Implementation Details.** To generate perturbations for each ICL prompt, we use a neighborhood size of $\sigma = 0.1$ and generate local perturbation neighborhoods $\mathcal{N}_x$ for each test sample **x**. We sample $n_x = 10,000$ points sampled for each neighborhood, where the values for $\sigma$ and $n_x$ were chosen to give an equal number of samples for each class, whenever possible. We present perturbations in two main formats: as the raw perturbed inputs alongside their corresponding outputs (shown in the Sec. 3.1 and 3.2 templates); or as the change between each perturbed input and the test sample, and the corresponding change in output (shown in Sec. 3.3). The second approach significantly aids the LLM in discerning the most important features (see Fig. 11), providing only the changes relative to the test sample, and bypassing the LLM's need to internally compute these differences. As a result, the consistent value of the original test point becomes irrelevant, and this clearer, relational view allows the LLM to focus directly on variations in input and output. Note that both of these formats are absent from Sec. 3.4, which uses test samples directly and does not compute perturbations.

For the LLMs, we use OpenAI's text generation API with a temperature of $\tau = 0$ for our main experiments. To evaluate the LLM explanations, we extract and process its answers to identify the top-$k$ most important features. We first save each LLM query's reply to a text file and use a script to

extract the features. We added explicit instructions like "... *provide your answer as a feature name on the last line. Do not provide any further details on the last line.*" to ensure reliable parsing of LLM outputs. In rare cases, the LLM won't follow our requested response format or it replies with "*I don't have enough information to determine the most important features.*" See Appendix 6.1 for further details.

## 4.2 RESULTS

Next, we discuss experimental results that answer key questions highlighted at the beginning of this section about LLMs as post hoc explainers (Q1-Q5).

**1) LLMs can generate faithful explanations.** We compare our proposed prompting-based LLM explanation strategies to existing post hoc explainers on the task of identifying important features for understanding ANN (Fig. 2) and LR (Fig. 3) model predictions across four real-world datasets (see Table 2). For the ANN model, the LLM-based explanations perform on par with the gradient-based methods (despite having white-box access to the underlying black-box model) and LIME (that approximates model behavior using a surrogate linear model). In particular, our proposed prompting strategies perform better than ITG, SHAP, a Random baseline, and a 16-sample version of LIME, namely $LIME_{16}$, which is analogous to the number of ICL samples used in the LLM prompts. We observe that LLM explanations, on average, achieve 51.74% lower PGU and 163.40% higher PGI than ITG, SHAP, and Random baseline for larger datasets (more number of features) like Adult and Credit compared to 25.56% lower PGU and 22.86% higher PGI for Blood and Recidivism datasets. While our prompting strategies achieve competitive PGU and PGI scores among themselves across different datasets for ANN models, the Instruction-based ICL strategy, on average across datasets, achieves higher FA and RA scores for the LR model (Fig. 3). We find that gradient-based methods and LIME achieve almost perfect scores on FA and RA metrics as they are able to get accurate model gradients and approximate the model behavior with high precision. Interestingly, the LLM-based explanations perform better than ITG, SHAP, and Random baseline methods, even for a linear model.

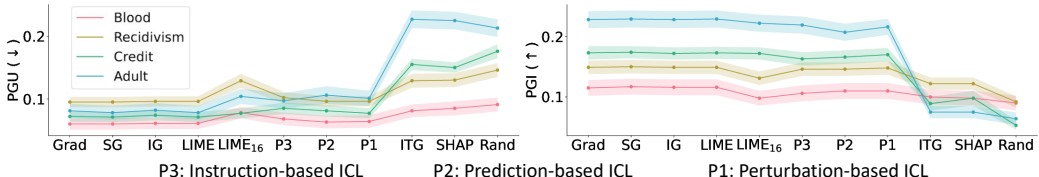

Figure 2: PGU and PGI scores of explanations generated using post hoc methods and LLMs (Instruction-based, Prediction-based, and Perturbation-based ICL prompting strategies) for an ANN model. On average, across four datasets, we find that LLM-based explanations perform on par with gradient-based and LIME methods and outperform $LIME_{16}$, ITG, and SHAP methods.

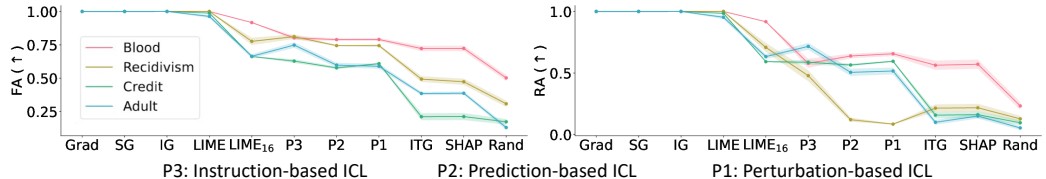

Figure 3: FA and RA scores of explanations generated using post hoc methods and LLMs (Instruction-based, Prediction-based, and Perturbation-based ICL prompting strategies) for an LR model. On average, across four datasets, we find that gradient-based methods and the LIME method (with 1000 samples) outperform all other methods and Instruction-based ICL explanations outperform other two prompting strategies across all datasets.

**2) LLM-augmented explainers achieve similar faithfulness to their vanilla counterparts.** We evaluate the faithfulness of the explanations generated using the Explanation-based ICL prompting strategy. Our results show that LLMs generate explanations that achieve faithfulness performance on par with those generated using state-of-the-art post hoc explanation methods for LR and large ANN predictive models across all four datasets (Fig. 4; see Table 3 for complete results) and four evaluation metrics. We demonstrate that very few in-context examples (here, $n_{ICL} = 4$) are sufficient to make the LLM mimic the behavior of any post hoc explainer and generate faithful explanations,

suggesting the effectiveness of LLMs as an explanation method. Interestingly, for low-performing explanation methods like ITG and SHAP, we find that explanations generated using their LLM counterparts achieve higher feature and rank agreement (Fig. 4) scores in the case of LR models, hinting that LLMs can use their internal knowledge to improve the faithfulness of explanations.

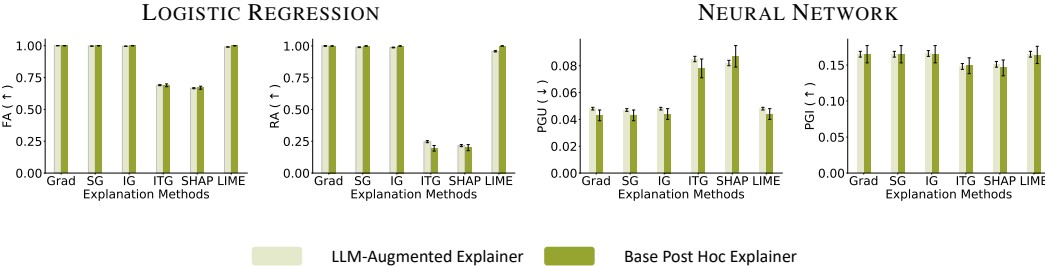

Figure 4: Faithfulness metrics on the Recidivism dataset for six post hoc explainers and their LLM-augmented counterparts for a given LR (left) and ANN (right) model. LLM-augmented explanations achieve on-par performance *w.r.t.* post hoc methods across all four metrics (see Table 3 for complete results on all other datasets).

**3) LLMs accurately identify the most important feature.** To demonstrate the LLM's capability in identifying the most important feature, we show the faithfulness performance of generated explanations across four datasets. Our results in Fig. 5 demonstrate the impact of different top-$k$ feature values on the faithfulness of explanations generated using our prompting strategies. We observe a steady decrease in RA scores (0.722 for top-$k$=1 vs. 0.446 for top-$k$=2 vs. 0.376 for top-$k$=4) across three datasets (Blood, Credit, and Adult) as the top-$k$ value increases. Interestingly, the RA value for top-$k$=1 for the Recidivism dataset is almost zero, though this can be attributed to the LLM's handling of the two primary features, whose LR coefficients have nearly identical magnitudes; the LLM generally places them both within the top two but, due to their similar importance, defaults to alphabetical order. However, when employing our Instruction-based ICL running-rank strategy, we find that the RA value rises from 0 to 0.5, highlighting the influence of nuanced prompts on the LLM's ranking mechanism. Further, we observe that LLMs, on average across four datasets and three prompting strategies, faithfully identify top-$k$=1 features with 72.19% FA score (see Fig. 12), and their faithfulness performance takes a hit for higher top-$k$ values. In the context of our 72.19% result, baseline methods' performances in identifying top-$k$=1 features are as follows: Random baseline (15%), SHAP (29.75%), ITG (29.5%), and LIME/IG/SG/Grad (100%) (see Tables 4-5).

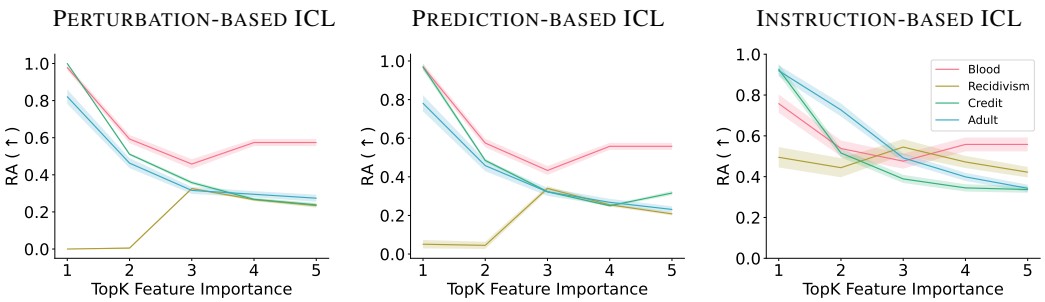

Figure 5: Effects of top-$k$ value on the RA metric using Perturbation-based, Prediction-based, and Instruction-based ICL prompting strategies. Shown are the results for three prompting strategies and four datasets using the LR model. On average, LLMs successfully achieve high scores in identifying the most important feature (top-$k$=1) and the performance decreases as we increase the top-$k$ value (see Fig. 12 for results on FA).

**4) GPT-3.5 vs. GPT-4.** An interesting question is how the reasoning capability of an LLM affects the faithfulness of the generated explanations. Hence, we compare the output explanations from GPT-3.5 and GPT-4 models to understand black-box model predictions. Results in Fig. 6-8 show that explanations generated using GPT-4, on average across four datasets, achieve higher faithfulness scores than explanations generated using the GPT-3.5 model. Across four prompting strategies, GPT-4, on average, obtains 4.53% higher FA and 48.01% higher RA scores than GPT-3.5 on explanations generated for the Adult dataset. We attribute this increase in performance of GPT-4 to its superior reasoning capabilities compared to the GPT-3.5 model (OpenAI, 2023). In Figure 6, we

find that Instruction-based ICL, on average across four datasets, outperforms the Perturbation-based ICL and Prediction-based ICL strategies on the RA metric. Further, our results in Fig. 8 show that the faithfulness performance of GPT-4 and GPT-3.5 are on par with each other when evaluated using our Explanation-based ICL strategy, which highlights that both models are capable of emulating the behavior of a post hoc explainer by looking at a few input, output, and explanation examples.

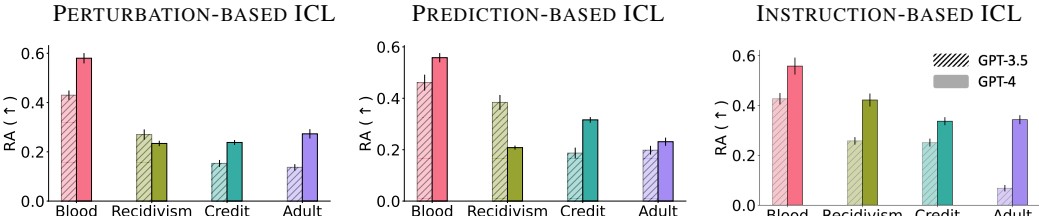

Figure 6: RA faithfulness metric of explanations generated using Perturbation-based ICL, Prediction-based ICL, and Instruction-based ICL prompting strategies on four real-world datasets. Explanations from GPT-4, on average, achieve higher RA scores than their GPT-3.5 counterparts (see Figures 8-9 for similar plots on Feature Agreement metric and Explanation-based ICL strategy).

**5) Ablation Study.** We conduct ablations on several components of the prompting strategies, namely the number of ICL samples, perturbation format, and temperature values. Results show that our choice of hyperparameter values is important for the prompting techniques to generate faithful post hoc explanations (Figs. 7,10). Our ablation on the number of ICL samples (Fig. 7) shows that fewer and larger numbers of ICL samples are not beneficial for LLMs to generate post hoc explanations. While fewer ICL samples provide insufficient information to the LLM to approximate the predictive behavior of the underlying ML model, a large number of ICL samples increases the input context, where the LLM struggles to retrieve relevant information from longer prompts, resulting in a decrease in the faithfulness of the explanations generated by LLMs. In contrast to LIME, the faithfulness of LLM explanations deteriorates upon increasing the number of ICL samples (analogous to the neighborhood of a given test sample). Across all four prompting strategies, we observe a drop in FA, RA, and PGI scores as we increase the number of ICL samples to 64. Further, our ablation on the temperature parameter of the LLMs shows that the faithfulness performance of the explanations does not change much across different values of temperature (see Appendix Fig. 10). Finally, results in Fig. 11 show that our prompting strategies achieve higher faithfulness when using the difference between the perturbed and test sample as input in the ICL sample.

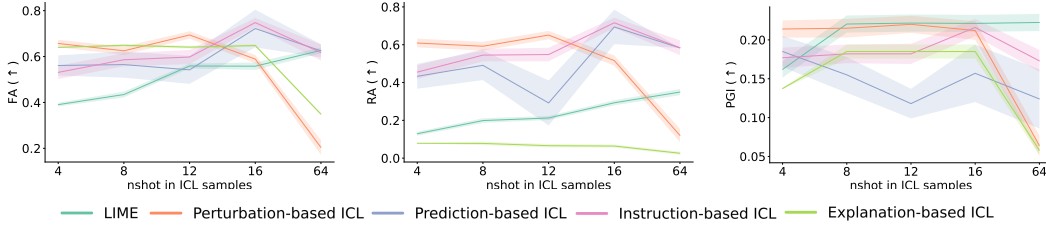

Figure 7: FA, RA, and PGI Performance of LIME and all four proposed prompting strategies as we increase the number of ICL samples (analogous to neighborhood samples in LIME) for the LR model trained on the Adult dataset. In contrast to LIME, the faithfulness of LLM explanations across different metrics decreases for a higher number of ICL samples, likely due to the limited capabilities of LLM for longer prompt length.

## 5 CONCLUSION

We introduce and explore the potential of using state-of-the-art LLMs as post hoc explainers. To this end, we propose four prompting strategies — Perturbation-based ICL, Prediction-based ICL, Instruction-based ICL, and Explanation-based ICL— with varying levels of information about the local neighborhood of a test sample to generate explanations using LLMs for black-box model predictions. We conducted several experiments to evaluate LLM-generated explanations using four benchmark datasets. Our results across different prompting strategies highlight that LLMs can generate faithful explanations and consistently outperform methods like ITG and SHAP. Our work paves the way for several exciting future directions in explainable artificial intelligence (XAI) to explore LLM-based explanation frameworks.

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

## 6 APPENDIX: ADDITIONAL RESULTS AND EXPERIMENTAL DETAILS

### 6.1 ADDITIONAL EXPERIMENTAL DETAILS

The median number of occurrences for cases where the LLM didn't follow our requested response format or it replies with *"I don't have enough information to determine the most important features"* is 3 for Perturbation-based ICL, 0.5 for Prediction-based ICL, and 0 for Explanation-based ICL. We use the LLM's top-$k$ features to calculate explanation faithfulness using four evaluation metrics. For calculating PGU and PGI metrics, we use perturbation mean $\mu_{PG}=0$, standard deviation $\sigma_{PG}=0.1$, and the number of perturbed samples $m_{PG}=10,000$. We follow the default hyperparameters from OpenXAI for generating explanations from standard post hoc explainers.

**Metrics.** We follow Agarwal et al. (2022) and used their evaluation metrics in our work. Below, we provide their respective definitions.

*a) Feature Agreement (FA)* metric computes the fraction of top-$K$ features that are common between a given post hoc explanation and the corresponding ground truth explanation.

*b) Rank Agreement (RA)* metric measures the fraction of top-$K$ features that are not only common between a given post hoc explanation and the corresponding ground truth explanation, but also have the same position in the respective rank orders.

*c) Prediction Gap on Important feature perturbation (PGI)* metric measures the difference in prediction probability that results from perturbing the features deemed as influential by a given post hoc explanation.

*d) Prediction Gap on Unimportant feature perturbation (PGU)* which measures the difference in prediction probability that results from perturbing the features deemed as unimportant by a given post hoc explanation.

For a given instance $\mathbf{x}$, we first obtain the prediction probability $\hat{y}$ output by the underlying model $f$, *i.e.*, $\hat{y} = f(\mathbf{x})$. Let $e_{\mathbf{x}}$ be an explanation for the model prediction of $\mathbf{x}$. In the case of PGU, we then generate a perturbed instance $\mathbf{x}'$ in the local neighborhood of $\mathbf{x}$ by holding the top-$k$ features constant, and slightly perturbing the values of all the other features by adding a small amount of Gaussian noise. In the case of PGI, we generate a perturbed instance $\mathbf{x}'$ in the local neighborhood of $\mathbf{x}$ by slightly perturbing the values of the top-$k$ features by adding a small amount of Gaussian noise and holding all the other features constant. Finally, we compute the expected value of the prediction difference between the original and perturbed instances as:

$$\text{PGI}(\mathbf{x}, f, e_{\mathbf{x}}, k) = \mathbb{E}_{\mathbf{x}' \sim \text{perturb}(\mathbf{x}, e_{\mathbf{x}}, \text{top-}K)}[|\hat{y} - f(\mathbf{x}')|], \tag{1}$$

$$\text{PGU}(\mathbf{x}, f, e_{\mathbf{x}}, k) = \mathbb{E}_{\mathbf{x}' \sim \text{perturb}(\mathbf{x}, e_{\mathbf{x}}, \text{non top-}K)}[|\hat{y} - f(\mathbf{x}')|], \tag{2}$$

where perturb($\cdot$) returns the noisy versions of $\mathbf{x}$ as described above.

**Hyperparameters for XAI methods.** Below, we provide the values for all hyperparameters of the explanation methods used in our experiments.

**a) LIME.** kernel_width = 0.75; std_LIME = 0.1; mode = 'tabular'; sample_around_instance = True; n_samples_LIME = 1000 or 16; discretize_continuous = False

**b) Grad.** absolute_value = True

**c) Smooth grad.** n_samples_SG = 100; std_SG = 0.005

**d) Integrated gradients.** method = 'gausslegendre'; multiply_by_inputs = False; n_steps = 50

**e) SHAP.** n_samples = 500

### 6.2 ADDITIONAL RESULTS

Here, we include additional and detailed results of the experiments discussed in Sec. 4.

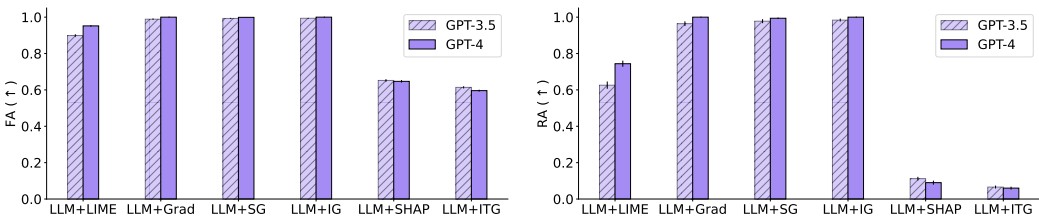

Figure 8: FA and RA metric performances for six LLM-augmented post hoc explainers (Sec. 3.4) when generating explanations for a given LR model using GPT-3.5 vs. GPT-4. Explanations from GPT-4, on average, outperform those generated using GPT-3.5 on both metrics on the Adult dataset.

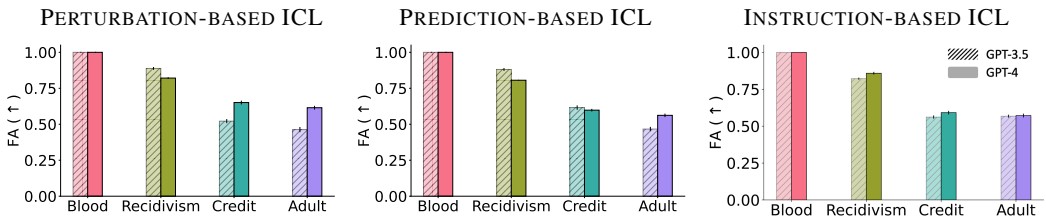

Figure 9: FA metric performances of explanations generated using Perturbation-based ICL, Prediction-based ICL, and Instruction-based ICL prompting strategies on four real-world datasets. Explanations from GPT-4, on average, achieve higher FA scores than their GPT-3.5 counterparts.

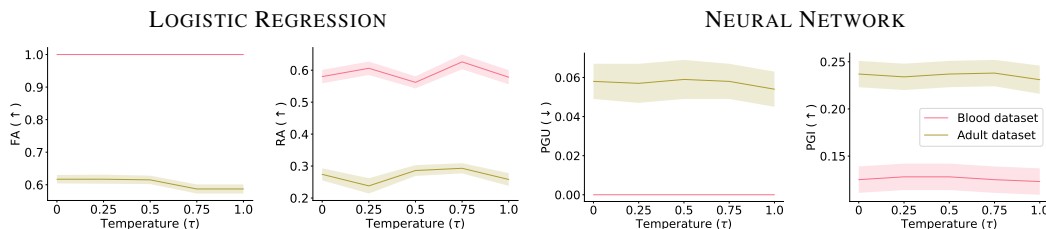

Figure 10: Metric performances for explanations generated using LLMs for different temperatures ($\tau$) with a Logistic Regression model (left) and a Neural Network (right) model. LLM-based explanations perform almost consistently across different temperature values, but LLMs will more often reply along the lines of *"not enough information to determine the most important features,"* for higher temperatures.

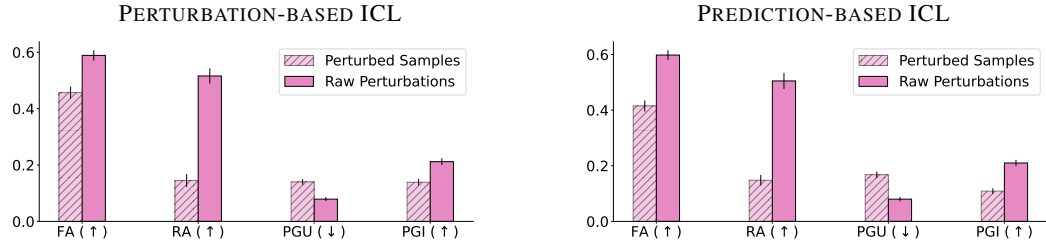

Figure 11: Faithfulness performance of explanations generated using Perturbation-based ICL (left) and Prediction-based ICL (right) on using perturbed samples vs difference between perturbed samples and the input sample (raw perturbations) in the ICL prompts for LR models trained on the Adult dataset. Across both prompting strategies, we find that using ICL samples using the raw perturbation style results in significantly better faithfulness performance across all four metrics.

Table 1: **Results of the machine learning models trained on four datasets.** Shown are the accuracy of the LR and ANN models trained the datasets. The best performance is bolded.

| Dataset | LR | ANN |
|---|---|---|
| Blood | **70.59%** | 64.71% |
| Recidivism | 76.90% | 76.90% |
| Default Credit | 87.37% | **88.34%** |
| Adult | 77.37% | **80.11%** |

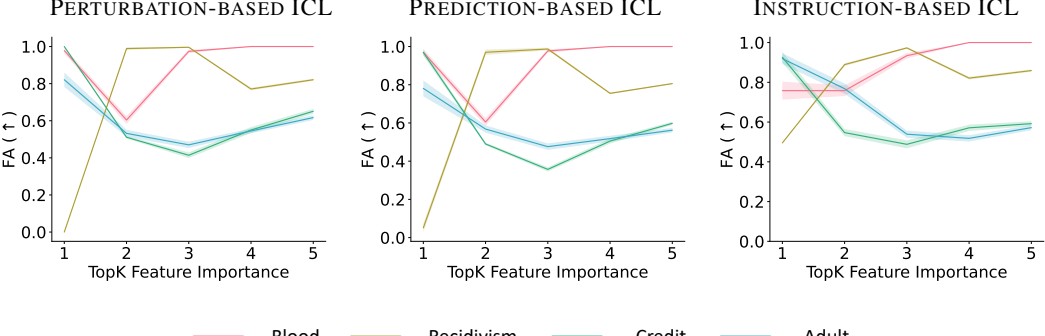

Figure 12: Effects of top-$k$ value on the FA explanation faithfulness metric when using Perturbation-based ICL, Prediction-based ICL, and Instruction-based ICL prompting strategies. Shown are the results for three prompting strategies and four datasets using the LR model. On average, LLMs successfully achieve high scores in identifying the most important feature (top-$k = 1$) and the performance decreases as we increase the top-$k$ value. For the Blood and Recidivism datasets, FA increases for top-$k \geq 4$ because they have four and six features in their dataset, respectively.

Table 2: Here we provide the average and standard error faithfulness metric values of explanations calculated across 100 instances in the test set. The results are generated using Perturbation-based ICL, Prediction-based ICL, Instruction-based ICL, six post hoc explanation methods, and a random baseline. For the LLM methods, we queried the LLM for the top-$k$ = 5 ($k$ = 4 for Blood) most important features and calculated each metric's area under the curve (AUC) for $k$ = 3 (where the AUC is calculated from $k$ = 1 to $k$ = 3). This will help us better understand the model's (Logistic Regression and Artificial Neural Network) predictions trained on four datasets. Arrows ($\uparrow$, $\downarrow$) indicate the direction of better performance.

| Dataset | Method | LR | | | | ANN | |
|---|---|---|---|---|---|---|---|
| | | FA ($\uparrow$) | RA ($\uparrow$) | PGU ($\downarrow$) | PGI ($\uparrow$) | PGU ($\downarrow$) | PGI ($\uparrow$) |
| Blood | Grad | 1.000±0.000 | 1.000±0.000 | 0.010±0.000 | 0.042±0.000 | 0.060±0.009 | 0.115±0.013 |
| | SG | 1.000±0.000 | 1.000±0.000 | 0.010±0.000 | 0.042±0.000 | 0.060±0.009 | 0.115±0.013 |
| | IG | 1.000±0.000 | 1.000±0.000 | 0.010±0.000 | 0.042±0.000 | 0.061±0.009 | 0.116±0.013 |
| | ITG | 0.722±0.019 | 0.563±0.037 | 0.019±0.001 | 0.037±0.001 | 0.081±0.010 | 0.100±0.012 |
| | SHAP | 0.723±0.020 | 0.556±0.037 | 0.019±0.001 | 0.036±0.001 | 0.085±0.011 | 0.098±0.012 |
| | LIME | 1.000±0.000 | 1.000±0.000 | 0.010±0.000 | 0.042±0.000 | 0.061±0.009 | 0.116±0.013 |
| | Random | 0.502±0.022 | 0.232±0.032 | 0.029±0.001 | 0.026±0.001 | 0.091±0.011 | 0.090±0.012 |
| | Perturbation-based ICL | 0.790±0.011 | 0.656±0.018 | 0.015±0.000 | 0.041±0.001 | 0.064±0.010 | 0.110±0.013 |
| | Prediction-based ICL | 0.789±0.009 | 0.638±0.018 | 0.014±0.000 | 0.041±0.000 | 0.063±0.010 | 0.110±0.013 |
| | Instruction-based ICL | 0.802±0.015 | 0.578±0.037 | 0.014±0.000 | 0.040±0.001 | 0.068±0.010 | 0.106±0.013 |
| Recidivism | Grad | 1.000±0.000 | 1.000±0.000 | 0.059±0.003 | 0.106±0.005 | 0.095±0.008 | 0.149±0.011 |
| | SG | 1.000±0.000 | 1.000±0.000 | 0.059±0.003 | 0.106±0.005 | 0.095±0.008 | 0.149±0.011 |
| | IG | 1.000±0.000 | 1.000±0.000 | 0.059±0.003 | 0.106±0.005 | 0.096±0.008 | 0.149±0.011 |
| | ITG | 0.493±0.021 | 0.214±0.030 | 0.090±0.005 | 0.078±0.004 | 0.129±0.011 | 0.122±0.010 |
| | SHAP | 0.473±0.023 | 0.217±0.032 | 0.092±0.005 | 0.076±0.004 | 0.130±0.011 | 0.122±0.010 |
| | LIME | 1.000±0.000 | 1.000±0.000 | 0.059±0.003 | 0.106±0.005 | 0.096±0.008 | 0.149±0.011 |
| | Random | 0.308±0.023 | 0.127±0.024 | 0.101±0.005 | 0.063±0.005 | 0.146±0.011 | 0.092±0.009 |
| | Perturbation-based ICL | 0.744±0.004 | 0.084±0.003 | 0.060±0.003 | 0.104±0.005 | 0.096±0.008 | 0.148±0.011 |
| | Prediction-based ICL | 0.744±0.008 | 0.120±0.017 | 0.061±0.003 | 0.103±0.005 | 0.096±0.008 | 0.146±0.011 |
| | Instruction-based ICL | 0.811±0.017 | 0.478±0.044 | 0.063±0.003 | 0.103±0.005 | 0.102±0.009 | 0.146±0.011 |
| Adult | Grad | 0.999±0.001 | 0.999±0.001 | 0.056±0.006 | 0.221±0.011 | 0.081±0.011 | 0.228±0.014 |
| | SG | 0.999±0.001 | 0.999±0.001 | 0.056±0.006 | 0.221±0.011 | 0.080±0.011 | 0.227±0.014 |
| | IG | 1.000±0.000 | 1.000±0.000 | 0.056±0.006 | 0.221±0.011 | 0.082±0.011 | 0.228±0.014 |
| | ITG | 0.385±0.012 | 0.099±0.019 | 0.215±0.011 | 0.061±0.007 | 0.227±0.014 | 0.075±0.010 |
| | SHAP | 0.387±0.012 | 0.150±0.020 | 0.215±0.011 | 0.061±0.007 | 0.225±0.014 | 0.075±0.010 |
| | LIME | 0.963±0.012 | 0.953±0.015 | 0.056±0.006 | 0.221±0.011 | 0.078±0.011 | 0.229±0.014 |
| | Random | 0.130±0.017 | 0.053±0.015 | 0.198±0.012 | 0.054±0.008 | 0.213±0.014 | 0.064±0.010 |
| | Perturbation-based ICL | 0.589±0.018 | 0.516±0.027 | 0.079±0.007 | 0.212±0.012 | 0.101±0.012 | 0.216±0.013 |
| | Prediction-based ICL | 0.598±0.017 | 0.505±0.029 | 0.080±0.008 | 0.210±0.011 | 0.106±0.014 | 0.207±0.014 |
| | Instruction-based ICL | 0.748±0.020 | 0.716±0.027 | 0.069±0.007 | 0.217±0.011 | 0.097±0.012 | 0.219±0.014 |
| Default Credit | Grad | 1.000±0.000 | 1.000±0.000 | 0.065±0.005 | 0.195±0.009 | 0.072±0.008 | 0.173±0.011 |
| | SG | 1.000±0.000 | 1.000±0.000 | 0.065±0.005 | 0.195±0.009 | 0.072±0.008 | 0.172±0.011 |
| | IG | 1.000±0.000 | 1.000±0.000 | 0.065±0.005 | 0.195±0.009 | 0.074±0.008 | 0.172±0.010 |
| | ITG | 0.211±0.026 | 0.157±0.026 | 0.150±0.006 | 0.106±0.012 | 0.155±0.009 | 0.089±0.011 |
| | SHAP | 0.212±0.026 | 0.161±0.026 | 0.150±0.006 | 0.107±0.012 | 0.150±0.008 | 0.098±0.012 |
| | LIME | 0.988±0.005 | 0.985±0.007 | 0.065±0.005 | 0.195±0.009 | 0.071±0.008 | 0.173±0.010 |
| | Random | 0.173±0.020 | 0.095±0.020 | 0.185±0.010 | 0.054±0.006 | 0.176±0.011 | 0.053±0.007 |
| | Perturbation-based ICL | 0.609±0.006 | 0.595±0.006 | 0.077±0.006 | 0.192±0.009 | 0.077±0.008 | 0.170±0.011 |
| | Prediction-based ICL | 0.577±0.009 | 0.565±0.010 | 0.080±0.007 | 0.189±0.009 | 0.081±0.009 | 0.166±0.011 |
| | Instruction-based ICL | 0.628±0.014 | 0.587±0.020 | 0.080±0.007 | 0.188±0.010 | 0.085±0.009 | 0.163±0.011 |

Table 3: Results of explanations generated using Explanation-based ICL and six post hoc explanation methods for understanding model (Logistic Regression and Artificial Neural Network) predictions trained on three datasets. Shown are average and standard error metric values computed across 100 test samples. Arrows (↑, ↓) indicate the direction of better performance. Evaluation metrics were computed for the top-$k$, $k$ being set to the number of features in each respective dataset.

| Dataset | Method | LR | | | | ANN | |
| --- | --- | --- | --- | --- | --- | --- | --- |
| | | FA (↑) | RA (↑) | PGU (↓) | PGI (↑) | PGU (↓) | PGI (↑) |
| Blood | LLM-Lime | 0.708±0.006 | 0.465±0.009 | 0.013±0.000 | 0.041±0.001 | 0.074±0.009 | 0.099±0.012 |
| | Lime | 1.000±0.000 | 1.000±0.000 | 0.008±0.000 | 0.043±0.000 | 0.044±0.006 | 0.121±0.013 |
| | LLM-Grad | 0.997±0.003 | 0.996±0.004 | 0.008±0.000 | 0.043±0.000 | 0.058±0.009 | 0.116±0.012 |
| | Grad | 1.000±0.000 | 1.000±0.000 | 0.008±0.000 | 0.043±0.000 | 0.044±0.006 | 0.120±0.013 |
| | LLM-SG | 0.990±0.006 | 0.983±0.011 | 0.008±0.000 | 0.043±0.000 | 0.055±0.008 | 0.116±0.012 |
| | SG | 1.000±0.000 | 1.000±0.000 | 0.008±0.000 | 0.043±0.000 | 0.044±0.006 | 0.120±0.013 |
| | LLM-IG | 0.989±0.005 | 0.982±0.009 | 0.008±0.000 | 0.043±0.000 | 0.046±0.007 | 0.120±0.013 |
| | IG | 1.000±0.000 | 1.000±0.000 | 0.008±0.000 | 0.043±0.000 | 0.044±0.006 | 0.120±0.013 |
| | LLM-Shap | 0.684±0.013 | 0.401±0.025 | 0.020±0.001 | 0.034±0.001 | 0.069±0.009 | 0.102±0.012 |
| | Shap | 0.773±0.014 | 0.516±0.033 | 0.015±0.001 | 0.038±0.001 | 0.066±0.009 | 0.107±0.012 |
| | LLM-ITG | 0.702±0.013 | 0.387±0.029 | 0.017±0.001 | 0.036±0.001 | 0.069±0.010 | 0.105±0.012 |
| | ITG | 0.774±0.014 | 0.532±0.034 | 0.014±0.001 | 0.038±0.001 | 0.063±0.008 | 0.108±0.012 |
| Recidivism | LLM-Lime | 0.990±0.001 | 0.958±0.005 | 0.029±0.001 | 0.115±0.002 | 0.048±0.001 | 0.165±0.004 |
| | Lime | 1.000±0.000 | 1.000±0.000 | 0.029±0.002 | 0.116±0.006 | 0.044±0.004 | 0.164±0.012 |
| | LLM-Grad | 0.997±0.001 | 0.990±0.003 | 0.029±0.001 | 0.115±0.002 | 0.048±0.001 | 0.165±0.004 |
| | Grad | 1.000±0.000 | 1.000±0.000 | 0.029±0.002 | 0.116±0.006 | 0.043±0.004 | 0.165±0.012 |
| | LLM-SG | 0.997±0.001 | 0.990±0.003 | 0.029±0.001 | 0.115±0.002 | 0.047±0.001 | 0.165±0.004 |
| | SG | 1.000±0.000 | 1.000±0.000 | 0.029±0.002 | 0.116±0.006 | 0.043±0.004 | 0.165±0.012 |
| | LLM-IG | 0.996±0.001 | 0.988±0.003 | 0.029±0.001 | 0.115±0.002 | 0.048±0.001 | 0.166±0.004 |
| | IG | 1.000±0.000 | 1.000±0.000 | 0.029±0.002 | 0.116±0.006 | 0.044±0.004 | 0.165±0.012 |
| | LLM-Shap | 0.666±0.004 | 0.216±0.008 | 0.057±0.001 | 0.098±0.002 | 0.082±0.002 | 0.151±0.004 |
| | Shap | 0.670±0.012 | 0.200±0.024 | 0.058±0.003 | 0.099±0.005 | 0.087±0.008 | 0.146±0.011 |
| | LLM-ITG | 0.690±0.004 | 0.247±0.008 | 0.056±0.001 | 0.099±0.002 | 0.085±0.002 | 0.148±0.004 |
| | ITG | 0.689±0.011 | 0.195±0.022 | 0.056±0.003 | 0.100±0.005 | 0.078±0.007 | 0.149±0.011 |
| Adult | LLM-Lime | 0.909±0.001 | 0.632±0.005 | 0.023±0.001 | 0.222±0.003 | 0.035±0.002 | 0.230±0.004 |
| | Lime | 0.907±0.005 | 0.743±0.017 | 0.018±0.002 | 0.224±0.011 | 0.029±0.005 | 0.235±0.014 |
| | LLM-Grad | 0.938±0.000 | 0.801±0.001 | 0.022±0.001 | 0.223±0.003 | 0.035±0.002 | 0.230±0.004 |
| | Grad | 0.999±0.001 | 0.997±0.003 | 0.018±0.002 | 0.224±0.011 | 0.029±0.004 | 0.234±0.014 |
| | LLM-SG | 0.938±0.000 | 0.802±0.001 | 0.022±0.001 | 0.223±0.003 | 0.035±0.002 | 0.230±0.004 |
| | SG | 0.999±0.001 | 0.997±0.003 | 0.018±0.002 | 0.224±0.011 | 0.029±0.004 | 0.234±0.014 |
| | LLM-IG | 0.938±0.000 | 0.804±0.000 | 0.022±0.001 | 0.223±0.003 | 0.033±0.002 | 0.231±0.004 |
| | IG | 1.000±0.000 | 1.000±0.000 | 0.018±0.002 | 0.224±0.011 | 0.031±0.005 | 0.235±0.014 |
| | LLM-Shap | 0.676±0.002 | 0.069±0.003 | 0.109±0.002 | 0.148±0.003 | 0.123±0.003 | 0.153±0.004 |
| | Shap | 0.662±0.007 | 0.107±0.012 | 0.139±0.009 | 0.127±0.009 | 0.144±0.011 | 0.149±0.013 |
| | LLM-ITG | 0.665±0.002 | 0.039±0.002 | 0.107±0.002 | 0.150±0.003 | 0.132±0.003 | 0.146±0.004 |
| | ITG | 0.627±0.006 | 0.068±0.010 | 0.175±0.010 | 0.099±0.009 | 0.170±0.011 | 0.130±0.013 |
| Default Credit | LLM-Lime | 0.954±0.001 | 0.787±0.003 | 0.030±0.001 | 0.189±0.003 | 0.042±0.002 | 0.178±0.003 |
| | Lime | 0.977±0.004 | 0.878±0.015 | 0.030±0.003 | 0.201±0.009 | 0.037±0.004 | 0.186±0.010 |
| | LLM-Grad | 0.984±0.000 | 0.896±0.001 | 0.029±0.001 | 0.189±0.003 | 0.042±0.002 | 0.178±0.003 |
| | Grad | 1.000±0.000 | 1.000±0.000 | 0.030±0.003 | 0.201±0.009 | 0.038±0.005 | 0.185±0.011 |
| | LLM-SG | 0.984±0.000 | 0.897±0.000 | 0.029±0.001 | 0.189±0.003 | 0.072±0.003 | 0.165±0.003 |
| | SG | 1.000±0.000 | 1.000±0.000 | 0.030±0.003 | 0.201±0.009 | 0.037±0.004 | 0.185±0.011 |
| | LLM-IG | 0.984±0.000 | 0.896±0.001 | 0.029±0.001 | 0.189±0.003 | 0.041±0.002 | 0.179±0.003 |
| | IG | 1.000±0.000 | 1.000±0.000 | 0.030±0.003 | 0.201±0.009 | 0.041±0.005 | 0.185±0.010 |
| | LLM-Shap | 0.543±0.003 | 0.067±0.004 | 0.088±0.002 | 0.140±0.003 | 0.094±0.003 | 0.126±0.003 |
| | Shap | 0.525±0.009 | 0.086±0.012 | 0.088±0.005 | 0.163±0.010 | 0.091±0.006 | 0.146±0.011 |
| | LLM-ITG | 0.526±0.003 | 0.052±0.003 | 0.088±0.002 | 0.139±0.003 | 0.091±0.002 | 0.129±0.003 |
| | ITG | 0.516±0.010 | 0.076±0.012 | 0.086±0.005 | 0.165±0.010 | 0.084±0.006 | 0.152±0.010 |

Table 4: Shown are the faithfulness scores for the most important feature value, top-$k = 1$, identified by existing post hoc explanation methods as well as the three LLM methods which generated explanations from GPT-4 across four datasets and the ANN model.

| Method | Recidivism | | Adult | | Credit | | Blood | |
|---|---|---|---|---|---|---|---|---|
| | PGU ($\downarrow$) | PGI ($\uparrow$) | PGU ($\downarrow$) | PGI ($\uparrow$) | PGU ($\downarrow$) | PGI ($\uparrow$) | PGU ($\downarrow$) | PGI ($\uparrow$) |
| Grad | $0.147_{\pm0.011}$ | $0.117_{\pm0.010}$ | $0.103_{\pm0.013}$ | $0.224_{\pm0.014}$ | $0.085_{\pm0.009}$ | $0.166_{\pm0.010}$ | $0.087_{\pm0.012}$ | $0.103_{\pm0.012}$ |
| SG | $0.146_{\pm0.011}$ | $0.117_{\pm0.010}$ | $0.103_{\pm0.013}$ | $0.224_{\pm0.014}$ | $0.084_{\pm0.009}$ | $0.167_{\pm0.010}$ | $0.087_{\pm0.012}$ | $0.102_{\pm0.012}$ |
| IG | $0.147_{\pm0.011}$ | $0.116_{\pm0.010}$ | $0.103_{\pm0.013}$ | $0.225_{\pm0.014}$ | $0.085_{\pm0.009}$ | $0.167_{\pm0.010}$ | $0.087_{\pm0.012}$ | $0.103_{\pm0.012}$ |
| ITG | $0.154_{\pm0.012}$ | $0.084_{\pm0.009}$ | $0.232_{\pm0.014}$ | $0.056_{\pm0.009}$ | $0.181_{\pm0.010}$ | $0.057_{\pm0.009}$ | $0.103_{\pm0.012}$ | $0.083_{\pm0.012}$ |
| SHAP | $0.152_{\pm0.012}$ | $0.092_{\pm0.009}$ | $0.231_{\pm0.014}$ | $0.047_{\pm0.008}$ | $0.169_{\pm0.009}$ | $0.076_{\pm0.011}$ | $0.104_{\pm0.012}$ | $0.083_{\pm0.012}$ |
| LIME | $0.147_{\pm0.011}$ | $0.116_{\pm0.010}$ | $0.104_{\pm0.013}$ | $0.225_{\pm0.014}$ | $0.084_{\pm0.009}$ | $0.167_{\pm0.010}$ | $0.087_{\pm0.012}$ | $0.103_{\pm0.012}$ |
| Random | $0.163_{\pm0.012}$ | $0.062_{\pm0.009}$ | $0.228_{\pm0.014}$ | $0.031_{\pm0.008}$ | $0.187_{\pm0.011}$ | $0.033_{\pm0.006}$ | $0.115_{\pm0.012}$ | $0.067_{\pm0.010}$ |
| Sec. 3.1 | $0.146_{\pm0.012}$ | $0.114_{\pm0.010}$ | $0.142_{\pm0.015}$ | $0.179_{\pm0.015}$ | $0.083_{\pm0.009}$ | $0.166_{\pm0.011}$ | $0.085_{\pm0.012}$ | $0.099_{\pm0.013}$ |
| Sec. 3.2 | $0.145_{\pm0.012}$ | $0.113_{\pm0.010}$ | $0.137_{\pm0.015}$ | $0.172_{\pm0.016}$ | $0.087_{\pm0.009}$ | $0.162_{\pm0.011}$ | $0.085_{\pm0.012}$ | $0.097_{\pm0.012}$ |
| Sec. 3.3 | $0.149_{\pm0.011}$ | $0.113_{\pm0.010}$ | $0.121_{\pm0.014}$ | $0.202_{\pm0.014}$ | $0.097_{\pm0.010}$ | $0.151_{\pm0.011}$ | $0.094_{\pm0.012}$ | $0.084_{\pm0.012}$ |

Table 5: Shown are the faithfulness scores for the most important feature value, top-$k = 1$, identified by existing post hoc explanation methods as well as the three LLM methods which generated explanations from GPT-4 across four datasets and the LR model. (Since FA = RA for top-$k = 1$, we omit RA to prevent redundancy).

| Method | Recidivism | | Adult | | Credit | | Blood | |
|---|---|---|---|---|---|---|---|---|
| | FA ($\uparrow$) | PGU ($\downarrow$) | FA ($\uparrow$) | PGU ($\downarrow$) | FA ($\uparrow$) | PGU ($\downarrow$) | FA ($\uparrow$) | PGU ($\downarrow$) |
| Grad | $1.000_{\pm0.000}$ | $0.096_{\pm0.005}$ | $1.000_{\pm0.000}$ | $0.073_{\pm0.007}$ | $1.000_{\pm0.000}$ | $0.081_{\pm0.006}$ | $1.000_{\pm0.000}$ | $0.020_{\pm0.000}$ |
| SG | $1.000_{\pm0.000}$ | $0.095_{\pm0.005}$ | $1.000_{\pm0.000}$ | $0.073_{\pm0.007}$ | $1.000_{\pm0.000}$ | $0.081_{\pm0.006}$ | $1.000_{\pm0.000}$ | $0.020_{\pm0.000}$ |
| IG | $1.000_{\pm0.000}$ | $0.096_{\pm0.005}$ | $1.000_{\pm0.000}$ | $0.073_{\pm0.007}$ | $1.000_{\pm0.000}$ | $0.081_{\pm0.006}$ | $1.000_{\pm0.000}$ | $0.020_{\pm0.000}$ |
| ITG | $0.190_{\pm0.039}$ | $0.108_{\pm0.006}$ | $0.020_{\pm0.014}$ | $0.221_{\pm0.011}$ | $0.270_{\pm0.044}$ | $0.163_{\pm0.007}$ | $0.700_{\pm0.046}$ | $0.026_{\pm0.001}$ |
| SHAP | $0.210_{\pm0.041}$ | $0.108_{\pm0.006}$ | $0.020_{\pm0.014}$ | $0.221_{\pm0.011}$ | $0.270_{\pm0.044}$ | $0.163_{\pm0.007}$ | $0.700_{\pm0.046}$ | $0.026_{\pm0.001}$ |
| LIME | $1.000_{\pm0.000}$ | $0.096_{\pm0.005}$ | $0.990_{\pm0.010}$ | $0.221_{\pm0.011}$ | $1.000_{\pm0.000}$ | $0.081_{\pm0.006}$ | $1.000_{\pm0.000}$ | $0.020_{\pm0.000}$ |
| Random | $0.130_{\pm0.034}$ | $0.113_{\pm0.006}$ | $0.060_{\pm0.024}$ | $0.214_{\pm0.011}$ | $0.070_{\pm0.026}$ | $0.195_{\pm0.010}$ | $0.190_{\pm0.039}$ | $0.038_{\pm0.001}$ |
| Sec. 3.1 | $0.000_{\pm0.000}$ | $0.101_{\pm0.005}$ | $0.821_{\pm0.039}$ | $0.101_{\pm0.010}$ | $1.000_{\pm0.000}$ | $0.081_{\pm0.007}$ | $0.978_{\pm0.015}$ | $0.020_{\pm0.000}$ |
| Sec. 3.2 | $0.051_{\pm0.022}$ | $0.101_{\pm0.005}$ | $0.781_{\pm0.042}$ | $0.109_{\pm0.010}$ | $0.969_{\pm0.017}$ | $0.084_{\pm0.007}$ | $0.970_{\pm0.017}$ | $0.020_{\pm0.000}$ |
| Sec. 3.3 | $0.490_{\pm0.050}$ | $0.098_{\pm0.005}$ | $0.919_{\pm0.027}$ | $0.086_{\pm0.009}$ | $0.926_{\pm0.027}$ | $0.090_{\pm0.007}$ | $0.758_{\pm0.045}$ | $0.025_{\pm0.001}$ |

Table 6: Run time in seconds across 100 samples for explanations generated using LLMs and other post hoc explanation methods.

| Method | LR | | | | ANN | | | | |
|---|---|---|---|---|---|---|---|---|---|
| | COMPAS | Blood | Adult | Credit | COMPAS | Blood | Adult | Credit | Mean runtime (in secs) |
| Grad | 0.183 | 0.001 | 0.002 | 0.001 | 0.003 | 0.001 | 0.002 | 0.002 | 0.024 |
| SG | 0.174 | 0.121 | 0.124 | 0.123 | 0.134 | 0.127 | 0.131 | 0.128 | 0.133 |
| IG | 0.044 | 0.043 | 0.043 | 0.043 | 0.047 | 0.045 | 0.047 | 0.046 | 0.045 |
| ITG | 0.001 | 0.001 | 0.001 | 0.001 | 0.001 | 0.001 | 0.002 | 0.001 | 0.001 |
| SHAP | 8.93 | 9.064 | 9.151 | 8.996 | 11.21 | 11.143 | 11.165 | 11.077 | 10.092 |
| LIME | 2.922 | 1.482 | 0.407 | 0.398 | 3.051 | 1.574 | 0.476 | 0.456 | 1.346 |
| LLM | 1732 | 1668 | 1418 | 1624 | 1578 | 1313 | 1349 | 1723 | 1550 |

