# OpenReview forum: "Are Large Language Models Post Hoc Explainers?"
_ICLR.cc/2024/Conference — Submitted to ICLR 2024_

### Official Review · Reviewer_nbwF · 2023-10-16

**Soundness:** 2 fair
**Presentation:** 3 good
**Contribution:** 2 fair
**Rating:** 3
**Confidence:** 4

**Summary:**

This paper aims to explain the black-box models' output by in-context learning on large language models (LLMs). To achieve this, the authors transformed the input into sentences and proposed four prompting strategies to generate different instructions for LLMs, using LLMs to extract top-k important features to explain the black-box model. The authors compare their faithfulness of explanation with other baselines, showing competitive results.

**Strengths:**

1. This paper is well-written and easy to understand.
2. The proposed method is easy to reproduce.
3. The authors provide enough experimental data to support their claim.

**Weaknesses:**

The soundness of this paper is poor. The authors treat LLMs as a principal component analysis model, use them to fit the data distribution and find the top-k most important features with different prompts as the explanations for black-box models. This is not a guaranteed process because it is unclear how LLMs fit the data distribution inside the prompt, not to mention how LLMs "understand" the data distribution and further provide a faithful explanation from the perspective of data. In fact, the "logical thinking skill"[1], the ability to process math problems[1], and the instruction-following ability of LLMs[2] are poor or remain unclear; even the order of the input will affect the output of a LLM[3].

To maximize the power of LLM, a better way is to post-hoc explain the model's output from the perspective of "natural language", like [4], which is easy to understand and easy to evaluate. Using language as output, the faithfulness of explanation can be easily evaluated by human annotators intuitively. Another way is to let LLMs use tools (e.g., use Python to code) to enhance the extra ability of LLMs and further obtain a guaranteed faithful explanation for a black-box model.

[1] Song et al. NLPBench: Evaluating Large Language Models on Solving NLP Problems. Arxiv 2023.

[2] Zeng et al. Evaluating Large Language Models at Evaluating Instruction Following. Arxiv 2023.

[3] Pouya et al. Large language models sensitivity to the order of options in multiple-choice questions. Arxiv 2023.

[4] Menon et al. Visual Classification via Description from Large Language Models. ICLR 2023

**Questions:**

N/A

---

> ### Author Response · Authors · 2023-11-19
> **Rebuttal for Reviewer nbwF**
>
> We thank the reviewer for acknowledging the reproducibility and experiments of our work. We greatly appreciate your feedback on solidifying the contributions of our work and address your concerns below.
>
> **Motivation for using LLMs as a post hoc explainer**
>
> We understand your concern about the use of LLMs as a post hoc explainer. Our choice to use Large Language Models (LLMs) for this task was **driven by our desire to explore their potential beyond conventional language processing applications.** The **novelty** of our approach lies in its ability **to integrate contextual understanding and in-context learning (ICL) capabilities of LLMs** to provide richer, more nuanced explanations than what might be achievable through simpler methods. LLMs are useful for the following reasons:
>
>
> Existing methods often **lack the ability to offer detailed, human-like explanations.** LLMs can bridge this gap by generating explanations that are more understandable to humans, which is particularly valuable in fields where interpretability is crucial, such as healthcare or finance.
>
>
> Generating explanations using LLMs is not just about identifying important features but also about understanding the rationale behind these selections in a manner that simpler methods may not provide.
>
>
> The ability of LLMs to process and explain complex patterns in data can be particularly beneficial when dealing with intricate, high-dimensional datasets where traditional feature selection methods might struggle.
>
>
> We would like to note that the potential benefits in terms of the depth and quality of explanations justify the exploration of our study, where we intended to be a stepping stone in the direction of using LLMs as possible post hoc explainers.
>
>
>
> **To maximize the power of LLM, a better way is to post-hoc explain the model's output from the perspective of "natural language" …… Another way is to let LLMs use tools (e.g., use Python to code) to enhance the extra ability of LLMs and further obtain a guaranteed faithful explanation for a black-box model.**
>
> Thank you for your valuable feedback on our work. We appreciate your insights on using the LLMs in other explainability frameworks and would like to note that there are  inherent challenges in using Large Language Models (LLMs) as explainers. However, we would like to clarify that **our work aims to explore the potential of LLMs in a new XAI domain** — introducing the first framework to study the effectiveness of LLMs in explaining other predictive models trained on tabular datasets. Regarding the application of LLMs to numerical datasets, we believe this represents an innovative step in understanding the capabilities of LLMs beyond their traditional scope. While LLMs are primarily designed for language tasks, **their ability to abstract and generalize can potentially be leveraged in a variety of contexts, including numerical data interpretation.** Our paper extends the TabLLM work [1] that studies the application of LLMs to zero-shot and few-shot classification of tabular data.
>
> **In fact, the “logical thinking skill”, the ability to process math problems[1], and the instruction-following ability of LLMs are poor or remain unclear; even the order of the input will affect the output of a LLM**
>
> We agree with the reviewer that the current set of LLMs suffers from a range of problems, such as processing math problems, lack of instruction-following ability, and inability to process longer inputs. However, we would like to clarify that our exploration of LLMs does not expect them to solve extensive math problems to generate explanations. For instance, we specifically instruct the LLM to explain its reasoning (**see the “Instructions” section of the prompt templates on pages 4 and 5**) before reaching a decision on which features are most important to the model, and the LLMs follow the rule explicitly stated in the instruction (see Section 3.3). Moreover, recent works like Lightman et al. [2] show that LLMs show enhanced performance on mathematical reasoning tasks, which will improve with the progress of LLM research. Again, our **goal is to explore the potential of LLMs as a post hoc explainer**, and that like other explanation methods, is not without its limitations.
>
> We are very grateful to the reviewer for all their questions/concerns, as they have helped us improve our paper significantly. We tried to address all the reviewer suggestions and hope the reviewer considers increasing their score.
>
>
> **References**
>
> [1] Hegselmann, S., Buendia, A., Lang, H., Agrawal, M., Jiang, X., & Sontag, D. Tabllm: Few-shot classification of tabular data with large language models. In AISTATS, 2023.
>
> [2] Lightman, H., Kosaraju, V., Burda, Y., Edwards, H., Baker, B., Lee, T., ... & Cobbe, K. Let's Verify Step by Step. arXiv, 2023.

---

> > ### Author Response · Authors · 2023-11-22
> > **Looking forward to your response**
> >
> > Dear reviewer,
> >
> > Thank you again for your thoughtful feedback. Following your suggestions, we have provided justifications regarding the motivation of using LLMs as post hoc explainers, as well as the remaining points on maximizing LLM utility. We would love to hear your thoughts on our response. Please let us know if there is anything else we can do to address your comments.
> >
> > Best,
> >
> > Authors of *Are Large Language Models Post Hoc Explainers?*

---

### Official Review · Reviewer_e81j · 2023-10-22

**Soundness:** 2 fair
**Presentation:** 2 fair
**Contribution:** 2 fair
**Rating:** 5
**Confidence:** 3

**Summary:**

The authors investigate the potential of using LLMs as explainers of the external models’ behavior. To this end, the authors explore 4 strategies to quantify feature importance. Authors compare existing feature attribution algorithms to LLM-based explainers, and demonstrate comparable performance to existing algorithms.

**Strengths:**

1. The idea of using LLMs as general-purpose explainers is an interesting one that could be practical and useful if it works well. This paper does a good job at taking an initial stab to demonstrate the potential of such approaches.

2. Authors show that LLMs can employ existing feature attribution paradigms, such as perturbation-based feature attributions to replace existing algorithms such as LIME or Shap. If the LLMs can perform better than or for cheaper than the existing algorithms, these approaches could be effective in practice.

**Weaknesses:**

1. (Pareto Curve) Even if the LLMs are decisively better/worse than the existing explainability methods, the ultimate decision of the practitioner would also be based on how cheap/expensive it is to obtain the explanations. I would be interested to see an approximate cost-benefit tradeoff to have a better sense of whether the LLM-in-the-loop explainers are preferable in practice. The cost here could be with respect to $ compute and with respect to time.

2. Authors focus on relatively simple tabular datasets, which I personally think limits the impact of the methodology and results. While I understand that it’s not yet possible to focus on many different domains (e.g. vision seems not possible yet); I believe it would have been possible to use existing text classifiers, as perturbations could still be communicated to existing models via prompts. This would increase the practicality and the value of the evaluation, in my view.

3. (Lack of sufficient experimental details) I find that the presentation of the experiments could be significantly improved. As it stands, there are a lot of missing details in the experiments section or the Appendix, which makes me feel less confident about the reliability of the results. I detail several points below. I believe most of these points could easily be addressed in the rebuttal phase, and I will be happy to revise my assessment during the rebuttal.

- 3.1 Importantly, I believe the authors should define their evaluation metrics; e.g., for metrics like PGI, PGU, RA, FA the authors refer to earlier papers without explicitly defining what they are. The reader should ideally be able to see the metrics without navigating to different papers.

- 3.2 I cannot find the hyperparameters of the explainers or the rationale for picking them. In particular, it’s unclear if the performance may or may not be explained by a poor choice of hyperparameters, as even the rationale of the hyperparameter choices for existing algorithms (LIME, SHAP etc.) are not presented in the paper. Since most of the results are meaningful in a relative sense (compared to the baselines), this is an important point to clarify.

- 3.3 Similarly, for reproducibility purposes, the details around the models used should be better provided. E.g. it’s unclear what optimizer is used to train the models, with which learning rate, whether early stopping is applied, and so on. This would surely raise reproducibility issues for follow-up work unless addressed.

- 3.4 The authors describe the process of parsing the response as `We first save each LLM query’s reply to a text file and use a script to extract the features.` I believe further details are needed to better understand this process. Specifically, what is the existing parsing strategy? Are the responses always parseable? What fraction of the time they are not parseable?

- 3.5 The authors present ` LLMs accurately identify the most important feature` as a significant result (e.g. abstract `identify the most important feature with 72.19% accuracy,`), however for this specific task I do not see baselines. Why do the authors have baselines for faithfulness, but not for this specific task (apologies if I’m missing this and the result exists)? Specifically – how good are existing algorithms at identifying the most important feature?

- 3.6 There are claims I find unjustified. For instance, `The second approach significantly aids the LLM in discerning the most important features` – how can we claim this without any results in Page 6 under implementation details? If there is an experimental finding that supports this, please refer to the result.

4. I’m slightly confused about the insights we can draw from the experiments. Specifically, the authors propose 3 different explanation strategies that seem to perform reasonably similarly. I understand the overall message that LLMs have the potential to be used as explainers. However, the confusing part to me is there are 3 algorithms presented, and it’s hard to understand which one is better or when. I’d appreciate it if the authors could provide a concise discussion around this.

**Questions:**

1. How do the authors pick the hyperparameters for the baseline explainers?

2. Could the authors please explicitly define the metric they are using?

3. Could the authors please clarify the claim on Page 6 `The second approach significantly aids the LLM in discerning the most important features`?

4. Do the authors have any insights about the costs of the explanation methods, to inform practitioners about whether it is worth using LLMs in practice?

5. How well do the baselines perform in the most important feature identification tasks?

6. Is it possible to verify the effectiveness of these methods in text classification tasks?

---

> ### Author Response · Authors · 2023-11-18
> **Rebuttal for Reviewer e81j (Part 1/2)**
>
> We thank the reviewer for acknowledging the motivation and presentation of our work. We appreciate your helpful feedback and address all the mentioned questions/concerns below.
>
>
> **Compute and runtime for generating explanations using LLMs**
>
> Great point! We would like to clarify that, since submission, the **cost of GPT-4 has already reduced 10-fold** and inference times have gone down too. In response to the reviewers’ concern, we provide a detailed table below that compares the runtime for generating explanations using LLMs and existing post hoc explanation methods.
>
> | **Method** |   LR   |   LR  |   LR  |   LR   |   ANN  |   ANN  |   ANN  |   ANN  | **Mean runtime (in secs)** |
> |:----------:|:------:|:-----:|:-----:|:------:|:------:|:------:|:------:|:------:|:--------------------------:|
> |            | COMPAS | Blood | Adult | Credit | COMPAS | Blood  | Adult  | Credit |                            |
> | Grad        |  0.183 | 0.001 | 0.002 |  0.001 |  0.003 |  0.001 |  0.002 |  0.002 |            0.024           |
> | SG         |  0.174 | 0.121 | 0.124 |  0.123 |  0.134 |  0.127 |  0.131 |  0.128 |            0.133           |
> | IG         |  0.044 | 0.043 | 0.043 |  0.043 |  0.047 |  0.045 |  0.047 |  0.046 |            0.045           |
> | ITG        |  0.001 | 0.001 | 0.001 |  0.001 |  0.001 |  0.001 |  0.002 |  0.001 |            0.001           |
> | SHAP       |  8.93  | 9.064 | 9.151 |  8.996 |  11.21 | 11.143 | 11.165 | 11.077 |           10.092           |
> | LIME       |  2.922 | 1.482 | 0.407 |  0.398 |  3.051 |  1.574 |  0.476 |  0.456 |            1.346           |
> | LLM       | 1732 | 1668 | 1418 | 1624 | 1578 | 1313 | 1349 | 1723 | 1550 |
>
>
> From the above table, we show that the time taken by LLMs to generate explanations is greater than the other explanation methods. However, we would like to highlight that the runtime for generating explanations using LLMs by query OpenAI APIs depends on a range of factors, including time of the day, server/requests overload, rate limit imposed by OpenAI, etc. However, the above runtime is expected to go down as the query time for OpenAI APIs improves.
>
>
> **Lack of experimental details**
>
> We apologize for the lack of clarity and provide additional experimental details below.
> *PGI/PGU and FA/RA metrics*
>
> We have added a detailed definition of the four metrics in Section 6.1 of our revised manuscript.
>
>
> *Hyperparameters for explanation methods*
>
> We followed OpenXAI and used the standard hyperparameters for these explanation methods. In response to the reviewers’ feedback, we detail them below for reference and have added them to the revised manuscript.
>
> **LIME**
>
> kernel_width           = 0.75
>
> std_LIME               = 0.1
>
> mode                   = 'tabular'
>
> sample_around_instance = True
>
> n_samples_LIME         = 1000 or 16
>
> discretize_continuous  = False
>
> **grad**
>
> absolute_value = True
>
> **Smooth grad**
>
> n_samples_SG = 100
>
> std_SG       = 0.005
>
> **Integrated gradients**
>
> method             = 'gausslegendre'
>
> multiply_by_inputs = False
>
> n_steps            = 50
>
> **SHAP**
>
> n_samples = 500
>
> *Reproducibility - what learning rate used to train models etc*
>
> All our models are trained in PyTorch using cross entropy and a class weighted term to encourage underrepresented classes to contribute more to the loss than the most popular class.
>
> Optimizer: Adam
>
> LR: 0.001
>
> Epochs: 100
>
> Data normalization: minmax
>
> Batch size: 64
>
> We have shared all these details in the Jupyter notebook shared in the supplementary material (Notebooks/TrainModels.ipynb)
>
> *What is our parsing strategy and how many bad replies?*
>
> The GPT-4 and GPT-3.5 responses are parseable an average of **96.4%** and **85%** of the time, respectively. We improve parsing ability by encouraging the LLM’s replies to be nicely formatted. This is done by requesting the ranked features to be on the last line in descending order of importance with no other information present. The exact parsing details can be found in our Supplementary Materials code folder **llms > response.py**.
>
> In general, we use regex and string manipulation to identify the last line of the reply. We then extract key parts of the last line like **“is”**, **“:”**, and **“=”** which indicate that the ranked features will begin on the right side of the matched string. An example reply is as follows:
>
> *“Therefore, the top five most important features, ranked from most important to least important, are: G, A, F, H, I.*
>
> *Answer: G, A, F, H, I”*
>
> This will be parsed by splitting the last line by **:** into two parts, namely “Answer” and “G, A, F, H, I”. The latter is split by a comma and stripped of any white space to be finally placed into an array for further processing and faithfulness evaluation.
>
> **Top k = 1 for Post Hoc Explainers**
>
> In response to the reviewers’ feedback, we conducted a new experiment and added Figure 13 to show the top-1 performance of existing XAI methods.

---

> > ### Author Response · Authors · 2023-11-18
> > **Rebuttal for Reviewer e81j (Part 2/2)**
> >
> > **Unjustified: “The second approach significantly aids the LLM in discerning the most important features”**
> >
> > Thank you for pointing this out! We will add the correct reference to Figure 11 that backs up this claim.
> >
> > **What are the experimental insights? Why do we have 3 different strategies that perform roughly the same? Concise discussion.**
> >
> > Great point! We agree with the reviewer that the key takeaway of our study is that LLMs have the potential to be used as a post hoc explainer. The three prompting strategies presented in our work perform similarly at identifying the top k = 1, 2, 3, 4, and 5 features (**see Figure 5**). However, according to Figure 3 and Table 2, Instruction-based ICL is, on average, better at Feature Agreement and Rank Agreement than the other two methods, Perturbation-based and Prediction-based ICL. For instance, Instruction-based ICL outperforms the other two methods in terms of FA and RA for the LR model on the Adult dataset (see Table 2).
> >
> > To address your second concern about when to use one over another, it seems across the four datasets that **Instruction-based ICL performs better in terms of our metrics.** However, choosing which one to use is nuanced.
> >
> > 1. Perturbation-based ICL is best for a simpler and more cursory way to identify key features influencing predictions where deeper analysis may not be necessary.
> >
> > 2. Prediction-based ICL is ideal when the task requires analysis of both the predictive and explanatory components.
> >
> > 3. Instruction-based ICL is best when a detailed step-by-step analysis of each feature’s impact is necessary. This method provides a more comprehensive understanding of the model’s behavior and feature importance.
> >
> >
> > We have included most of the above clarifications and details in the revised version of the manuscript. We hope we addressed all your questions/concerns/comments adequately. In light of these clarifications, we kindly request you to consider increasing your score. We are very happy to answer any further questions.

---

> ### Comment · Reviewer_e81j · 2023-11-19
> **Response to Rebuttal**
>
> Thank you for your rebuttal and clarifications.
>
> > **Compute and Runtime**
>
> Thank you for this clarification in terms of the runtime, this is informative. Do we have a cost comparison too? Please correct me if I'm wrong, but the numbers suggest (to me) that it is i) Slower (albeit expected to get faster) ii) A lot more expensive (due to the API request costs). I think it would be good to acknowledge these limitations and be upfront in the paper (this does not diminish its value, but rather increases it with better transparency).
>
> Another question: Why are these details not in the paper?
>
> > **Experimental Details**
>
> Thank you for these clarifications. I'm not familiar with how OpenXAI picked these specific values -- is there a reason why the authors did not perform a hyperparameter sweep? From the compute time table above, it should not be too costly to quickly run a hyperparameter study, in my opinion.
>
> Overall, I'm still unclear why the authors do not share these experimental details in the main text / at least the appendix. Is there a specific reason? As a reviewer/reader of the paper, I would personally appreciate having these details at least in the appendix (and not in a notebook in the zip file). I don't think these important experimental details are to be found by readers in the lines of a jupyter notebook, neither the rebuttal page is the only place to make these clarifications.
>
> > **Experimental insights**
>
> - Thank you for your clarifications here. I do not see a Figure 13 in the paper, is it perhaps not updated?
> - I similarly do not see the discussion in the main text. Could you refer me to concretely where it is?
>
> I will revisit my evaluation after these clarifications.

---

> > ### Author Response · Authors · 2023-11-19
> > **Rebuttal response**
> >
> > Thank you for your quick response to our rebuttal response.
> >
> > **Compute and Runtime**
> >
> > Great point! In response to the reviewer’s feedback, we have added these runtime results in Table 6. Further, we did some cost analysis for our experiments using GPT-4 and have shared the estimates below for the Recisivism dataset (containing six features).
> >
> > i) For each sample in the Recidivism dataset and topk=5 explanation using the LLMs, we have 857 input tokens and 375 output tokens.
> >
> > ii) The OpenAI API cost for each prompt is **$0.03/1k** token and **$0.06/1k** token for the input and output, respectively.
> >
> > iii) Hence, the cost for one sample is = 0.857 x 0.03 + 0.375 x 0.06 = **$0.048**
> >
> > iv) Total cost for generating explanations for 100 samples of the recidivism dataset is **$4.80**
> >
> > We will add the cost estimate for prompting and generating explanations for all our datasets in Section 6 of our appendix. We apologize for not adding this detail in the initial version as we thought the cost information was mostly public (https://openai.com/pricing). However, we agree with the reviewer that this additional information will be useful to the XAI community.
> >
> > **Experimental details**
> >
> > We apologize for the lack of information in our initial draft. OpenXAI is a widely recognized and commonly used benchmark in the field of XAI. The primary reason we did not perform a hyperparameter sweep is to **align our results with the predictive models, datasets, and explanation methods present in the OpenXAI benchmark**, as it provides a standardized framework for evaluating and comparing different XAI methods, ensuring that our results are comparable and relevant to the broader research community. **We followed suit and adhered to a recognized benchmark** that would allow our work to be directly compared with other studies in the field. This choice was crucial for ensuring the validity and relevance of our findings, especially given the novelty of using LLMs for post hoc explanations in XAI.
> >
> > We didn’t share the hyperparameter and experimental details because we referred them from OpenXAI. However, we understand the reviewer’s point that adding these details will enhance the readability of our current draft. In light of the rebuttal discussions, **we included most of these details in Section 6.1 of the revised Appendix** and will add further details next.
> >
> > **Thank you for your clarifications here. I do not see a Figure 13 in the paper, is it perhaps not updated?**
> >
> > We sincerely apologize for this typo. We initially thought of adding a new figure but then ended up adding two new tables for these results (**see Tables 4-5 of the revised Appendix**).
> >
> > **I similarly do not see the discussion in the main text. Could you refer me to concretely where it is?**
> >
> > Please refer to the research question (3) on Page 8 of the revised manuscript. We have marked the additional text in blue for your reference.
> >
> > We hope these clarifications help in addressing your concerns and additional questions. Let us know if you have further queries.

---

> > > ### Comment · Reviewer_e81j · 2023-11-21
> > > **Response to authors**
> > >
> > > Thank you for the additional clarifications.
> > >
> > > > Choice of hyperparameters
> > >
> > > First, I did not review or at any point use OpenXAI and am not familiar with the rationale for picking 1 fixed set of hyperparameters, but I do not think we should evaluate the current paper based on choices from other papers. However, from practical experience, I know that the results could significantly change based on the choice of hyperparameters of explainers, across different use cases and datasets. e.g. LIME does k-lasso, and in higher dimensional cases one needs to have the k larger; how can we pick a fixed k across all experiments?
> > >
> > > > Top-k = 1 experiments (Tables 4 and 5)
> > >
> > > Thanks for pointing me to these. Am I interpreting the results correctly that there are methods better than LLMs? If yes, the message in the abstract `we observe that LLMs identify the most important feature with 72.19% accuracy, opening up new frontiers in explainable artificial intelligence (XAI) to explore LLM-based explanation frameworks` feels a bit misleading to me as I'm not sure why this particular result opens up new frontiers.
> > >
> > > Overall, I thank the authors for their rebuttal. While I still have some of the concerns outlined above, the authors also resolved some other concerns, such as reporting the compute/time/performance tradeoffs and expanding on the experimental details. I will be revising my score considering the above discussion.

---

> ### Author Response · Authors · 2023-11-21
> **Response to Reviewer e81j**
>
> We would like to thank the reviewer for their engagement and for reconsidering their initial score. These discussions have been helpful for us whilst revising our paper.
>
> **Choice of hyperparameters**
>
> This is a fair point regarding the rigidity we have assumed of the post hoc explainer hyperparameters. While we do believe it best to adhere to OpenXAI for means of comparison and benchmarking, this is an interesting discussion to have. It might well be the case that for even higher dimensional datasets than those used, picking one set of hyperparameters for LIME would not yield optimal results across all settings. We would like to make some quick follow-up points to address this in the context of our work:
>
> - We elected to use LIME with 1000 perturbations to mitigate any inconsistency or sub-optimality in its results.
> - More importantly, we also observe in the logistic regression setting that LIME achieves near-perfect FA/RA scores across datasets, as well as, in the neural network setting, performance on par with the other state-of-the-art methods that do not require hyperparameter tuning, i.e. gradients.
> - Lastly, the set-up we use evaluates explanations based on the *order* of top-k features (according to absolute value), and does not penalize based on *exact* feature importance values. We echo the reviewer's points that hyperparameter choices will affect the exact feature importance values returned by methods such as LIME. However, the overall *ranking* of the features would be affected to a lesser degree (evidenced by the previous point also) when averaged across the full dataset.
>
> We thank the reviewer again for making this point, as it is an important consideration for follow-up work where explanation metrics could be based on values rather than rankings.
>
> **Top-k = 1 experiments (Tables 4 and 5)**
>
> Thank you for reporting the misleading statement. We found that GPT-4 exhibits non-trivial performance for most important feature identification, and performs on par with state-of-the-art methods in certain settings like the Credit dataset (Tables 4 and 5). Comparing GPT-3.5 and GPT-4 (Figure 6) additionally demonstrates that LLMs are closing the performance gap to post hoc explainers. We deemed this a new frontier as high performance is achieved through prompting alone, a novel approach in our field.
>
> In light of this, we have updated the sentence in the abstract to be clearer, reading: `we observe that LLMs identify the most important feature with 72.19% accuracy, indicating promising avenues for further research into LLM-based explanation frameworks within explainable artificial intelligence (XAI).`
>
> We hope these points can help to address the reviewer's remaining concerns, and are happy to provide further clarifications if desired.

---

### Official Review · Reviewer_WeJ2 · 2023-11-01

**Soundness:** 1 poor
**Presentation:** 2 fair
**Contribution:** 1 poor
**Rating:** 3
**Confidence:** 2

**Summary:**

This paper proposes using language models to provide post-hoc explanations for other model decisions in four ways.

**Strengths:**

This paper addresses an important problem.

**Weaknesses:**

The critical weakness, in my view, is that the interpretability method itself is uninterpretable. When the task involves understanding language, this can be somewhat understood, like Bills et al.'s 2023 "Language models can explain neurons in language models." But this paper applies language models to ask them directly, "What are the most important features in determining the model's prediction?" And it does so on purely numerical datasets. It oversimplifies the task of interpretability, arbitrarily modeling logistic regression coefficients as fixed ground truths and using prediction gap -- but there are a million dramatically cheaper ways that they could have selected the most important features according to the same criteria.

Also, a notable limitation (if the more fundamental questions did not overshadow it) is that the models interpreted are all quite simple, the datasets are themselves simple, and more standard models for tabular data are not considered (but again, this is not the paper's primary limitation). Lastly, it seems presumptive to suggest that this approach is better than SHAP, at least without a deeper investigation into where this method outperforms it (and some discussion on why it ostensibly performs almost as poorly as randomly selecting features). These points are less important to me, but they are still worth raising.

**Questions:**

1) What do you mean when you say the logistic regression model has "one layer of size 16"?
2) Can you elaborate on the motivation for this work - why did you feel that a language model would be an appropriate tool here?
3) When would you use this approach instead of LIME, which consistently performed better?
4) Can you give some examples where SHAP performed worst according to your metrics? What were the values produced? What were the correct metrics?

---

> ### Author Response · Authors · 2023-11-18
> **Rebuttal for Reviewer WeJ2 (Part 1/2)**
>
> Thank you for your insightful questions. We are glad that you recognize the importance of this problem and we appreciate your feedback as they have helped us improve our paper significantly.  Below, we address all your questions/concerns.
>
> **Our method is uninterpretable**
>
> While we agree with the reviewer that the inner workings of LLMs are inherently complex and (currently) uninterpretable by nature, it is not the primary focus of our work and is an important point worth exploring in future work. Further, we would like to clarify that the explanations generated by the LLM using our prompting strategy are not *uninterpretable* or *fundamentally flawed*. For example, we specifically instruct the LLM to explain its reasoning (**see the “Instructions” section of the prompt templates on pages 4 and 5**) before reaching a decision on which features are most important to the model.
>
> We would like to note that identifying key features using LLMs is an interesting future direction as our current exploration shows that they achieve non-trivial performance when compared to some state-of-the-art explanation methods. Further, LLMs provide natural language explanations and a level of textual reasoning that is more accessible/interpretable to end users when explaining a model’s feature space. For instance, LIME performs a specific job that does not adapt to any specific patterns it observes in the perturbations, whereas LLMs are not limited in this respect, and have the capacity to adapt their explanation based on the patterns they see in the local neighborhood, which offers an explanation for why GPT-4 is more sample efficient than LIME with 16 input perturbations.
>
> We thank the reviewer for raising these points as we had not clearly articulated them in the original manuscript.
>
> **Comparison to Bills et al. 2023**
>
> Thank you for your valuable feedback on our paper. We appreciate your insights and the opportunity to address the concerns you raised regarding the interpretability of our proposed method.
>
> Firstly, we acknowledge your point about the interpretability challenge inherent in using Large Language Models (LLMs) as explainers. The complexity and opaqueness of these models do indeed raise valid concerns about the 'black-box' nature of the explanations they generate. However, **our work aims to explore the potential of LLMs in a new domain**, specifically as tools for post hoc explanation in XAI (Explainable Artificial Intelligence). Regarding the application of LLMs to numerical datasets, we believe this represents an innovative step in understanding the capabilities of LLMs beyond their traditional scope. While LLMs are primarily designed for language tasks, **their ability to abstract and generalize can potentially be leveraged in a variety of contexts, including numerical data interpretation.**
>
> With respect to the oversimplification of experimental setup in our approach to interpreting logistic regression coefficients. It was a deliberate design choice for initial experiments as we aimed to establish a baseline understanding of LLMs before moving on to more complex scenarios.
>
> **Motivation for using LLM instead of other cheaper alternatives**
>
> Thank you for your insightful critique regarding the cost-effectiveness of our approach in selecting the most important features. We understand your concern about there being more straightforward and less resource-intensive methods available for feature selection. Our choice to use Large Language Models (LLMs) for this task was **driven by our desire to explore their potential beyond conventional language processing applications.** The **novelty** of our approach lies in its ability **to integrate contextual understanding and in-context learning (ICL) capabilities of LLMs** to provide richer, more nuanced explanations than what might be achievable through simpler methods.
>
> While we agree that there are numerous simpler and more cost-effective ways to identify important features, LLMs are useful for the following reasons:
>
> 1. Existing methods often **lack the ability to offer detailed, human-like explanations.** LLMs can bridge this gap by generating explanations that are more understandable to humans, which is particularly valuable in fields where interpretability is crucial, such as healthcare or finance.
>
> 2. Generating explanations using LLMs is not just about identifying important features but also about understanding the rationale behind these selections in a manner that simpler methods may not provide.
>
> 3. The ability of LLMs to process and explain complex patterns in data can be particularly beneficial when dealing with intricate, high-dimensional datasets where traditional feature selection methods might struggle.

---

> > ### Author Response · Authors · 2023-11-18
> > **Rebuttal for Reviewer WeJ2 (Part 2/2)**
> >
> > We acknowledge that the cost and computational resources required for LLMs are significant. However, we believe that the potential benefits in terms of the depth and quality of explanations justify the exploration of this approach. Our work is intended to be a stepping stone in this direction, and we are actively exploring ways to optimize the efficiency and reduce the computational demands of our method.
> >
> > **more standard models for tabular data are not considered**
> >
> > We would like to clarify a potential misunderstanding here. The primary reason for selecting the specific tabular datasets in our study was to align with the datasets used in OpenXAI, a widely recognized and commonly used benchmark in the field of XAI. OpenXAI provides a standardized framework for evaluating and comparing different XAI methods, ensuring that our results are comparable and relevant to the broader research community. We followed suit and adhered to a recognized benchmark that would allow our work to be directly compared with other studies in the field. This choice was crucial for ensuring the validity and relevance of our findings, especially given the novelty of using LLMs for post hoc explanations in XAI. However, we acknowledge your point that incorporating a broader range of standard models for tabular data would have enriched our analysis, which remains part of our future work.
> >
> > **What do you mean when you say the logistic regression model has "one layer of size 16"?**
> >
> > We apologize for the lack of clarity. We use PyTorch to implement the LR and ANN models. The ANN has three hidden layers of size 64, 32, and 16, using ReLU for the hidden layers and Softmax for the output.
> >
> > **Can you give some examples where SHAP performed worst according to your metrics**
> >
> > In response to the reviewers’ question, we compare SHAP and LIME explanations generated for a sample from the Recidivism dataset trained using the LR model. The Recidivism dataset has six features (A, B, C, D, E, F).
> >
> > *SHAP explanation*
> >
> > The SHAP algorithm generates the feature ranking magnitudes as [0.2977, -0.0028,  0.0040, -0.1067,  0.00004, -0.0026] and the feature ranked by SHAP magnitude: ['A', 'D', 'C', 'B', 'F', 'E'] which represents **['age', 'c_charge_degree_F', 'length_of_stay', 'priors_count', 'race', 'sex_Female']** Intuitively, SHAP outputs individual feature magnitude values and they are generated using a somewhat game theory based algorithm.
> >
> > *LLM explanations*
> >
> > In contrast to SHAP and other existing methods, LLMs provide detailed reasoning of the model behavior.
> >
> > *Response from LLM:*
> >
> > Feature 'A':
> >
> > When 'A' increases, the output remains the same (0). When 'A' decreases, the output decreases (-1). This suggests a negative correlation between 'A' and the output.
> >
> > Importance: 80. 'A' seems to have a strong influence on the output, as a decrease in 'A' consistently leads to a decrease in the output.
> >
> > **Rank: A**
> >
> > Feature 'B':
> >
> > When 'B' increases, the output decreases (-1). When 'B' decreases, the output remains the same (0). This suggests a negative correlation between 'B' and the output.
> >
> > Importance: 90. 'B' seems to have a stronger influence on the output than 'A', as an increase in 'B' consistently leads to a decrease in the output.
> >
> > **Rank: BA**
> >
> > Feature 'C':
> >
> > When 'C' increases, the output decreases (-1). When 'C' decreases, the output remains the same (0). This suggests a negative correlation between 'C' and the output.
> >
> > Importance: 70. 'C' seems to have a slightly weaker influence on the output than 'A' and 'B', as changes in 'C' do not consistently lead to changes in the output.
> >
> > **Rank: BAC**
> >
> > …
> >
> > …
> >
> > …
> >
> > …
> >
> >
> > **Final Rank: BACDEF**
> >
> > Hence, LLMs provide extensive reasoning for the ranked features and identify the correct top 3 feature rank (ground truth for this sample was BACFDE)
> >
> > *Comparisons and Analysis*
> >
> > **Ground truth:** 1. Priors_count, 2. age, 3. length_of_stay
> >
> > **Top 3 SHAP explanations:** 1. age, 2. c_charge_degree_F, and 3. Length_of_stay
> >
> > **LLM explanations:** 1. Priors_count, 2. Age, 3. length_of_stay,
> >
> > LLM was able to perfectly match the explanations of the LR model SHAP only matched ‘age’ (in the wrong order) and length of stay in the correct position.
> >
> > We are very grateful to the reviewer for all their suggestions, as they have helped us improve our paper significantly. We tried to incorporate all the reviewer suggestions in our response. In light of these updates, we would kindly request the reviewer to consider increasing their score.

---

> ### Comment · Reviewer_WeJ2 · 2023-11-21
>
> Thank you for the response. I'd like to follow up on a few points.
>
> > Further, we would like to clarify that the explanations generated by the LLM using our prompting strategy are not uninterpretable or fundamentally flawed. For example, we specifically instruct the LLM to explain its reasoning (see the “Instructions” section of the prompt templates on pages 4 and 5) before reaching a decision on which features are most important to the model.
>
> This response seems to conflate interpretability and explainability -- I stand by my original point that LLM outputs are not interpretable. To the claim that they are explainable because the model outputs its reasoning, I again do not believe this is true because of the challenge of faithfulness. Namely, you do not know that the generated reasoning is reflective of the models' internal processing. Notably, faithfulness can be used in the context of reasoning and explanations, e.g. [1,2], and in the context of interpretability, e.g. [3], with slightly different meanings. Notably, just because your language model's generated explanations are faithful with respect to the explained model, that does *not* mean that they are faithful with respect to the language model. More concretely, if you had solved the faithfulness challenge in language models, that would be more than worthy of a paper of its own.
>
> > Our choice to use Large Language Models (LLMs) for this task was driven by our desire to explore their potential beyond conventional language processing applications... The ability of LLMs to process and explain complex patterns in data can be particularly beneficial when dealing with intricate, high-dimensional datasets where traditional feature selection methods might struggle.
>
> One question that still has not been answered is, why? What about language models makes them particularly well suited to this task? No results in this paper suggest that language models are better at dealing with high-dimensional data than other machine-learning approaches. You described my comment as "Motivation for using LLM instead of other cheaper alternatives," but I think "cheaper" misplaces my main concern: we are talking about a type of model that is fundamentally a black box and is well-known to hallucinate - there's no reason to trust that its explanations are correct, faithful, or generalize.
>
> > LLM was able to perfectly match the explanations of the LR model SHAP only matched ‘age’ (in the wrong order) and length of stay in the correct position.
>
> SHAP and LIME impose different priors, but I believe few people in the field would describe LIME as generally better than SHAP. More consistently matching LIME heuristically may suggest that the model's predictions are more local but not necessarily better. I think this additional context is useful, but as I noted in the original review, was not my primary concern.
>
> Taking all of this into account, I stand by my comments and my score.
>
> [1] "Faithful Reasoning Using Large Language Models" Creswell and Shanahan 2022
> [2] "Faithfulness Tests for Natural Language Explanations" Atanasova et al. 2023
> [3] "A Comparative Study of Faithfulness Metrics for Model Interpretability Methods" Chan et al. 2022

---

> > ### Author Response · Authors · 2023-11-22
> > **Response to Reviewer WeJ2**
> >
> > We would like to thank the reviewer for raising challenging questions regarding the legitimacy of using LLMs as post-hoc explainers. We sincerely appreciate your time in discussing our justifications.
> >
> > **On interpretability and faithfulness of explanations**
> >
> > The reviewer is correct about the distinction between interpretability and explainability; we apologize for our confusing usage where we intended to state that the explanations themselves could be interpreted. Regarding faithfulness, we should note that the purpose of our paper is specifically to assess the faithfulness of LLM explanations with respect to the explained model. We agree that this does not equate to faithfulness with respect to the LLM's internal processes, which is not the goal of our work. Rather, it is something we believe belongs squarely in future work, given the challenge associated there, as the reviewer rightly points out. However, we argue too that we cannot disregard high faithfulness scores on the explained model level based on uncertainty of internal faithfulness. Ideally, this is the first stepping stone towards explanations that are faithful in both respects. While LLMs have their shortcomings, they do not *always* hallucinate or reason incorrectly [1,2].
> >
> > **Why LLMs?**
> >
> > We apologize for the lack of clarity with respect to why we thought of exploring the use of LLMs as post hoc explainers. The key motivation for our exploration was the TabLLM work [1], which showed that state-of-the-art LLMs can achieve good zero-shot and few-shot classification performance on tabular data (some of which we used in our experiments; see Section 4.1). Regarding the utility of LLMs to high-dimensional datasets, we argue that results from recent works like TabLLM demonstrate an innovative step in understanding the capabilities of LLMs beyond their traditional scope, where despite being primarily designed for natural language tasks, their ability to abstract and generalize can potentially be leveraged in a variety of contexts, including numerical data understanding.
> >
> > Thank you again for engaging in discussion with us. We appreciate your time and efforts.
> >
> > [1] Hegselmann, S., Buendia, A., Lang, H., Agrawal, M., Jiang, X., & Sontag, D. Tabllm: Few-shot classification of tabular data with large language models. In AISTATS, 2023.
> >
> > [2] Lightman, H., Kosaraju, V., Burda, Y., Edwards, H., Baker, B., Lee, T., ... & Cobbe, K. Let's Verify Step by Step. arXiv, 2023.

---

### Meta-Review · Area_Chair_RY8q · 2023-12-10

**Metareview:**

The submission explores the idea of using language models for interpretability, by using LLMs for posthoc explanations of other models. Four methods are introduced for this problem. Reviewers agree that this is an interesting direction, and highly impactful if can be made to work convincingly. However, at the current stage, reviewers believe it falls beneath the bar for acceptance. In particular, only relatively simple models
 are interpreted, and reviewers believe that simpler and much cheaper methods may work just as well.

**Justification For Why Not Higher Score:**

Reviewers agree that this is a clear reject

**Justification For Why Not Lower Score:**

N/A

---

### Decision · Program_Chairs · 2024-01-16

Reject